



# HyLands 1.0: a Hybrid Landscape evolution model to simulate the impact of landslides and landslide-derived sediment on landscape evolution

Benjamin Campforts[1,2], Charles M. Shobe[1], Philippe Steer[3], Matthias Vanmaercke[4], Dimitri Lague[3], and Jean Braun[1,5]

[1]Helmholtz Centre Potsdam, GFZ German Research Centre for Geosciences, Potsdam, Germany
[2]Institute for Arctic and Alpine Research, University of Colorado at Boulder, Boulder, CO, USA
[3]Univ Rennes, CNRS, Géosciences Rennes – UMR 6118, 35000 Rennes, France
[4]Université de Liège, UR SPHERES, Département de Géographie, Clos Mercator 3, 4000 Liège, Belgium
[5]Institute of Earth and Environmental Science, Universität Potsdam, Potsdam, Germany

**Correspondence:** Benjamin Campforts (benjamin.campforts@gfz-potsdam.de)

**Abstract.** Landslides are the main source of sediment in most mountain ranges. Rivers then act as conveyor belts, evacuating landslide-derived sediment. Sediment dynamics are known to influence landscape evolution through interactions among landslide sediment delivery, fluvial transport, and river incision into bedrock. Sediment delivery and its interaction with river incision therefore control the pace of landscape evolution and mediate relationships among tectonics, climate, and erosion. Numerical landscape evolution models (LEMs) are well suited to study the interaction among these earth surface processes. They enable evaluation of a range of hypotheses at varying temporal and spatial scales. While many models have been used to study the dynamic interplay between tectonics, erosion and climate, the role of interactions between landslide-derived sediment and river incision has received much less attention. Here, we present HyLands, a hybrid landscape evolution model integrated within the Topo Toolbox Landscape Evolution Model (TTLEM) framework. The hybrid nature of the model lies in its capacity to simulate both erosion and deposition at any place in the landscape due to fluvial bedrock incision, sediment transport and rapid, stochastic mass wasting through landsliding. Fluvial sediment transport and bedrock incision are calculated using the recently developed Stream Power with Alluvium Conservation and Entrainment (SPACE) model. Therefore, rivers in HyLands can dynamically transition from detachment-limited to transport-limited, and from bedrock to bedrock-alluvial to fully alluviated states. Erosion and sediment production by landsliding is calculated using a Mohr-Coulomb stability analysis while landslide-derived sediment is routed and deposited using a multiple flow direction, non-linear deposition method. We describe and evaluate the HyLands 1.0 model using analytical solutions and observations. We first illustrate the functionality of HyLands to capture river dynamics ranging from detachment-limited to transport-limited configurations. Second, we apply the model to a portion of the Namche-Barwa massif in Eastern Tibet and compare simulated and observed landslide magnitude-frequency and area-volume scaling relationships. Finally, we illustrate the relevance of explicitly simulating landsliding and sediment dynamics over longer timescales for landscape evolution in general and river dynamics in particular. With HyLands we pro-



vide a new tool to understand both the long and short-term coupling between stochastic hillslope processes, river incision, and source-to-sink sediment dynamics.

# 1  Introduction

Landsliding is a highly effective erosional mechanism that dominates sediment mobilization rates in moderate to steep topographic settings (Hovius et al., 1997; Ouimet et al., 2007; Broeckx et al., 2020). Nonetheless, long term landscape evolution in non-glaciated settings is mainly controlled by the interplay between tectonic uplift and fluvial dynamics (Whipple and Tucker, 1999; Wobus et al., 2006). Fluvial channels in mountainous catchments play a dual role: they simultaneously incise into the bedrock and act as conveyor belts to carry eroded sediment out of the mountain range towards the ocean (Milliman and Meade,

1983). Through sediment evacuation and bedrock incision, fluvial incision lowers the base level for surrounding hillslopes, triggering hillslope failures. In turn, hillslope failure through mass wasting chokes the rivers with sediment and prevents further bedrock incision until landslide derived sediment has been evacuated from the system (Larsen and Montgomery, 2012; Ouimet et al., 2007; Korup et al., 2010; Shobe et al., 2016; Glade et al., 2019).

Unravelling the dynamic interplay between landslides and fluvial processes is key to understanding long-term landscape

evolution and the associated sediment dynamics in mountainous terrain (Egholm et al., 2013). Increased insight into the spatial distribution of landslides has resulted in improved landslide susceptibility assessments (Guzzetti et al., 2006), but processes regulating landslide rate assessments (Broeckx et al., 2020) and landslide-derived sediment dynamics remain less well understood (Hovius et al., 2011; Croissant et al., 2017, 2019; Zhang et al., 2019; Broeckx et al., 2020).

Numerical models are excellent tools to study relationships between processes regulating Earth surface dynamics and their

inter-dependencies over various temporal and spatial scales (Tucker and Hancock, 2010). The past twenty years have seen the development of a plethora of Landscape Evolution Models (LEMs), enabling studies of the interactions among climate, tectonics, and erosion. A crucial ingredient for any LEM is a fluvial erosion component regulating the way in which rivers transport sediment and incise into bedrock. Fluvial incision is controlled by both water and sediment cascading through river channels (Whipple et al., 2000; Hancock and Anderson, 2002; Turowski et al., 2007). Most existing LEMs simulate river

incision using one of two commonly used end member models to simulate fluvial dynamics (Armitage et al., 2018). In one approach, river incision is simulated assuming a detachment-limited configuration where erosion is constrained by the power to erode particles from the river bed and quantified using a scaling law between fluid stress and river incision rate (Seidl and Dietrich, 1992; Howard and Kerby, 1983; Campforts and Govers, 2015). In the other approach, river incision is simulated assuming a transport-limited configuration where erosion is constrained by the capacity of the river to carry sediment, where

the carrying capacity is a function of the fluid stress (Willgoose et al., 1991; Paola and Voller, 2005). These two formulations lead to similar outcomes in steady-state channels (where the river erosion rate equals the rock uplift rate), but noticeably different outcomes during transient river response to tectonic and climatic perturbations (Whipple and Tucker, 1999).

In real settings, however, even steep mountain channels undergoing long-term bedrock incision may experience bed cover by alluvial sediment. Further, over geologic time as tectonic and climatic forcings change, it is likely that any given channel





transitions between detachment-limited and transport-limited behavior. Such heterogeneous configurations require a model setup that can dynamically transition between detachment-limited and transport-limited regimes (e.g., Davy and Lague, 2009) and can simultaneously simulate fluvial sediment transport and river incision into bedrock (e.g., Lague, 2010). Recently, the SPACE (Stream Power with Alluvium Conservation and Entrainment) model approach was proposed to meet both of these needs (Shobe et al., 2017). Because SPACE is purely a river incision model, it does not simulate hillslope or mass wasting

processes. Additional model components are therefore needed to simulate the impact of mass wasting on landscape evolution and sediment dynamics.

To understand how landslides influence landscape evolution, Densmore et al. (1998) proposed an approach, adapted by others (e.g. Champel, 2002; Egholm et al., 2013), to integrate stochastic landslide dynamics in a numerical landscape evolution model. Densmore et al. (1998) assume that (i) all hillslopes behave as Mohr-Coulomb materials (Taylor, 1948), (ii) landslides

initialize in river channels (i.e. at the base of hillslopes) and (iii) landslide-derived sediment is spread under a constant slope, following the steepest downslope path. Assuming that landslides initialize only in fluvial channels makes it computationally easier to implement a landslide model because sediment pathways on hillslopes need not be calculated. However, it is not realistic to neglect landslides that might initiate away from river channels: it has been shown that a large portion of landslide-derived sediment is stored along flow paths on hillslopes, rather than being immediately delivered to river channels (Broeckx

et al., 2020; Hoffmann, 2015). The approach of Densmore et al. (1998) does not allow for landslide-derived sediment to be deposited and spread over hillslopes. Rather, landslide debris is spread as tongues of sediment filling up the river channel.

Other researchers have developed mechanistic models to simulate shallow landslide activity at the landscape scale (e.g. Montgomery and Dietrich, 1994; Claessens et al., 2007). Such models typically involve the explicit simulation of a soil layer and a coupled hydrologic model to calculate how changing pore water pressures trigger landslides (Van Asch et al., 1999;

Iverson, 2000; Baum et al., 2010). Although such mechanistic models are useful for assessing landslide hazards (e.g. to simulate landslide liquefaction associated with the Oso landslide, cfr. Iverson and George, 2016), they typically involve a range of geophysical processes and associated input parameters which are not always available at large spatial scales. This can make the more detailed models sensitive to equifinality (Beven and Freer, 2001). Moreover, deep-seated bedrock landslides, rather than shallow landslides, mobilize the largest volumes of sediment and therefore have the largest impact on landscape evolution

(Burbank, 2002; Dussauge et al., 2003; Jeandet et al., 2019; Korup et al., 2007; Broeckx et al., 2020).

In this paper, we present HyLands, a new Hybrid Landscape evolution model for simulating the interaction of landslide dynamics and fluvial processes. The model is intended to simulate Earth surface evolution at large spatial scales with a special focus on landsliding and the long-term effects of landslide-derived sediment. HyLands is integrated in the TTLEM 1.0 landscape evolution model (Campforts et al., 2017). Unlike the existing implementation of TTLEM, HyLands is a fully mass

conservative model where fluvial dynamics are modelled using the SPACE fluvial incision framework (Shobe et al., 2017) and hillslope-derived sediment fluxes are explicitly simulated. In this paper, we first describe the fluvial and landslide components of the HyLands model. We verify the fluvial model component by comparing model behavior against known analytical solutions. Subsequently, we evaluate the performance of the landslide module by applying HyLands to a selected region of the landslide-prone Namche-Barwa massif in Eastern Tibet. We show that HyLands reproduces observed landslide scaling





relationships. Next, we apply the model to a synthetic case to illustrate the potential of HyLands for studying the dynamic interaction between landslide activity and fluvial dynamics. We do this by evaluating how a steady-state landscape responds to an imposed pulse of landsliding activity. Finally, we discuss the current model limitations, future perspectives and a range of potential applications.

## 2   HyLands model description

HyLands is a Matlab model code building on the existing TopoToolbox Landscape Evolution Model (TTLEM, Campforts et al., 2017). It simulates changes in bedrock height and sediment thickness on a regular grid. The model is mass conservative; sediment produced by river incision and hillslope processes such as landsliding is explicitly simulated in the model. At every model iteration, the elevation of all grid cells is updated according to the following conservation statement for sediment and rock:

$$
\begin{aligned}
\frac{\partial \eta}{\partial t} =& \frac{\partial R}{\partial t} + \frac{\partial H}{\partial t} \\
=& U - E_{r_{fluv}} + \left( \frac{D_{s_{fluv}} - E_{s_{fluv}}}{1 - \phi_{sed}} \right) \\
& - E_{r_{hill}} + \left( \frac{D_{s_{hill}} - E_{s_{hill}}}{1 - \phi_{sed}} \right)
\end{aligned}
\tag{1}
$$

  where $\eta$ $[L]$ is the topographic elevation given by the sum of the bedrock elevation $R$ $[L]$ and the bed sediment thickness $H$ $[L]$. $U$ $[L/T]$ is the rock uplift rate and $\phi_{sed}$ is the bed sediment porosity. $E_{r_{fluv}}$ $[L/T]$ is the fluvial volumetric erosion flux

of bedrock per unit bed area, representing the amount of bedrock that is detached and entrained into the water column. $E_{s_{fluv}}$ $[L/T]$ is the fluvial volumetric entrainment flux of sediment per unit bed area and $D_{s_{fluv}}$ $[L/T]$ is the fluvial volumetric deposition flux of sediment per unit bed area. $E_{r_{hill}}$ $[L/T]$ is the volumetric flux of hillslope bedrock erosion (landslding) per unit bed area, representing the amount of bedrock that is detached. $E_{s_{hill}}$ $[L/T]$ is the volumetric entrainment flux of sediment erosion (produced by landsliding or creep) per unit bed area and $D_{s_{hill}}$ $[L/T]$ is the volumetric deposition flux of

hillslope-derived sediment per unit bed area.

### 2.1   River sediment transport and bedrock erosion

  HyLands uses the Stream Power with Alluvium Conservation and Entrainment (SPACE) river erosion model of Shobe et al. (2017). SPACE has two key advantages for the purposes of modeling river response to landslide sediment delivery. First, because of its derivation from the erosion-deposition family of models (e.g., Beaumont et al., 1992; Davy and Lague, 2009), it

can dynamically shift between detachment-limited (erosion is limited by the rate of sediment or bedrock detachment from the bed) and transport-limited (erosion is limited by the capacity of the flow to move detached sediment) behavior. Second, it can simulate the full continuum of possible river bed compositions from bare bedrock channels to mixed bedrock-alluvial channels to fully alluvial channels. This is accomplished by combining mass conservation of river bed sediment with a bedrock erosion





law to simultaneously solve for the time evolution of the bedrock and sediment surfaces. We implement the SPACE model
equations in the TTLEM MATLAB modeling framework. For a full overview of the SPACE model and comparison with other
models for coupled sediment and bedrock channel evolution, see Shobe et al. (2017).

### 2.1.1 Fluvial sediment and rock mass conservation

Conservation of sediment closely follows the erosion-deposition approach of Davy and Lague (2009), with the addition of terms
that represent the entrainment of detached bedrock in the water column (Fig. 1). The spatial change in volumetric sediment
flux $Q_{sfluv}$ $\left[L^3/T\right]$ per unit width $w$ $[L]$ is written as:

$$\frac{\partial \left(Q_{sfluv}/w\right)}{\partial x} = E_{s_{fluv}} + \left(1 - F_{f_{fluv}}\right) E_{r_{fluv}} - D_{s_{fluv}}. \tag{2}$$

where $F_{f_{fluv}}$ is a unitless fraction of fine fluvial sediment. The factor $1 - F_{f_{fluv}}$ $[-]$ represents the idea that a fraction of the
bedrock particles detached from the bed may be small enough to stay in permanent suspension, and therefore should not be
tracked as bed sediment.

### 2.1.2 Fluvial sediment entrainment, bedrock erosion, and sediment deposition

To evaluate the impact of landslide-derived sediment on landscape evolution, it is critical to have a model that simulates simul-
taneous sediment entrainment and bedrock erosion, and considers the influence of sediment cover on river erosion dynamics.
In the SPACE model, sediment entrainment and bedrock erosion may occur simultaneously. Further, the magnitude of each
process is set by the relative availability of sediment on the channel bed. SPACE accomplishes this by including the influence
of sediment layer on sediment and bedrock erosion rates.

Sediment entrainment and bedrock erosion are both governed by a unit stream power expression in which erosive power is
a function of water discharge per unit width $q$ $\left[L^2 T^{-1}\right]$ and local channel slope $S$ (e.g., Howard and Kerby, 1983; Whipple
and Tucker, 1999; Davy and Lague, 2009). The sediment erosion rate $E_{s_{fluv}}$ and the bedrock erosion rate $E_{r_{fluv}}$ are modified
by a term $H/H_*$ $[-]$ that encapsulates the ratio of bed sediment thickness $H$ $[L]$ to bedrock bed roughness $H_*$ $[L]$. High bed
sediment thickness or low bedrock surface roughness leads to a condition in which $H/H_*$ is large and little bedrock is exposed
to erosive flows. If bed sediment thickness is low or bedrock roughness is high, $H/H_*$ is small and most of the in-channel
bedrock is exposed to the flow.

SPACE assumes an exponential increase in sediment entrainment rate with $H/H_*$ and a concomitant exponential decrease
in bedrock erosion rate with $H/H_*$ (Fig. 2). Rates of sediment entrainment and bedrock erosion can therefore be written as
(assuming a negligible erosion threshold; see Shobe et al. (2017) for equations that relax this assumption):

$$E_{s_{fluv}} = K_s q S^n \left(1 - e^{-H/H_*}\right) \tag{3}$$

for sediment and

$$E_{r_{fluv}} = K_r q S^n e^{-H/H_*} \tag{4}$$



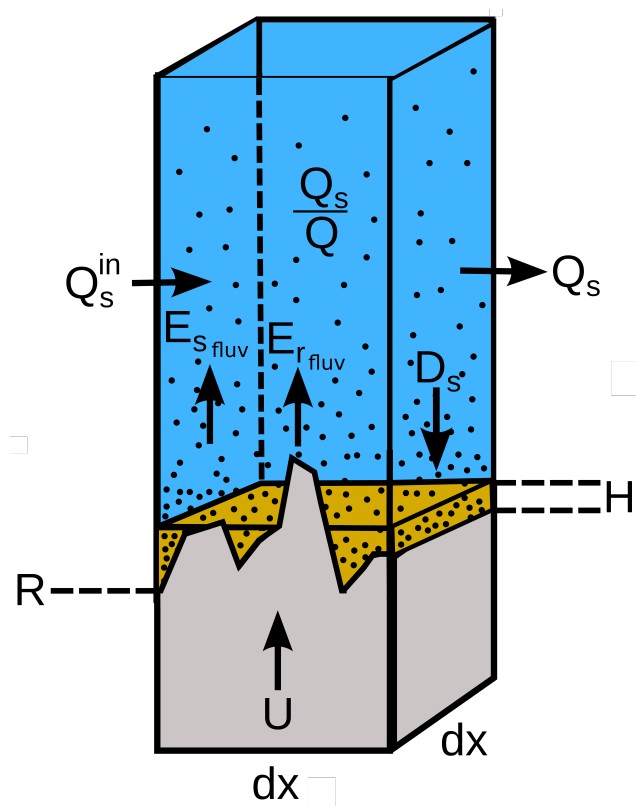

**Figure 1. Sketch of fluvial SPACE component.** Model setup and variable definitions for the SPACE bedrock-alluvial river erosion model. Reproduced from Shobe et al. (2017). Entrainment and deposition of sediment, as well as erosion of bedrock, can occur simultaneously. This approach allows channels to dynamically transition among bedrock, bedrock-alluvial, and fully alluviated states. At a given stream power, the relative rate of sediment entrainment $E_{s_{fluv}}$ and bedrock erosion $E_{r_{fluv}}$ is set by the ratio of sediment thickness $H$ to the bedrock roughness height $H_*$ (Fig. 2).

for bedrock. $K_s$ and $K_r$ $[L^{-1}]$ are erodibility constants for sediment and rock, respectively. $n$ is a constant set to 1 for all simulations in this paper, but that need not be 1 for the SPACE model in general. There are a variety of ways to compute water discharge $q$. We use the common approach of calculating discharge as a function of drainage area such that $q = k_q A^m$, where $m$ is a scaling exponent and $k_q$ is a coefficient subsumed into the fluvial erosion coefficients $K_s$ and $K_r$.

Sediment deposition is implemented similar to Davy and Lague (2009) such that the deposition flux depends on sediment flux $Q_{sfluv}$ divided by the volumetric water discharge $Q$ $[L^3/T]$ and the effective sediment settling velocity $V$ $[L/T]$:

$$D_s = \frac{Q_{sfluv}}{Q} V. \tag{5}$$

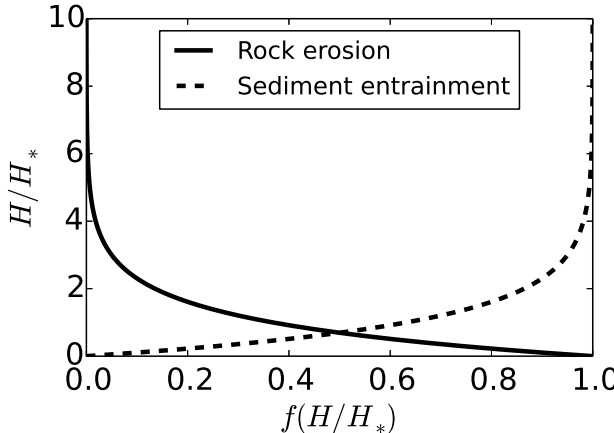

**Figure 2. Relative efficiency of fluvial sediment entrainment and bedrock erosion ($f(H/H_*)$) as a function of the ratio of sediment thickness to bedrock roughness $H/H_*$.** $f(H/H_*)$ depends only on the ratio $H/H_*$, and varies between 0 and 1, and indicates the proportion of total stream power used to erode bedrock (solid line) or sediment (dashed line). As sediment thickness $H$ increases relative to the bedrock roughness length scale $H_*$, the sediment entrainment rate factor approaches 1 and the bedrock erosion rate factor approaches 0 because the bed becomes composed entirely of sediment and no bedrock is exposed. As sediment thickness declines relative to the bedrock roughness length scale, the bedrock erosion rate increases exponentially because more bedrock is exposed and the sediment entrainment rate declines exponentially due to a lack of available sediment. This approach implements the "cover effect," in which the presence of sediment reduces bedrock erosion rates, but does not incorporate the "tools effect," in which mobile sediment enhances bedrock erosion. Reproduced from Shobe et al. (2017).

## 2.2 Landsliding

HyLands treats landslide erosion and deposition deterministically, but uses a stochastic approach to calculating landslide occurrence. HyLands simulates deep-seated gravitational landslides eroding simultaneously the sediment layer and the bedrock (erosion terms $E_{s_{hill}}$ and $E_{r_{hill}}$ respectively in Eq. 1). We assume that both the rock and the sediment layer behave as Mohr-Coulomb materials. In its current form, HyLands does not simulate shallow landslides where failure geometry is imposed by the depth and angle of soil-rock transitions. Landslide initiation does not involve a preceding triggering event (e.g. an earthquake) but is simulated using a probabilistic approach.

### 2.2.1 Landslide erosion

Following Densmore et al. (1998), we simulate landslide erosion using the Culmann theory for slope stability. Culmann (1875) proposes that hillslope failure will occur on the plane where the shear stress is balanced by the sliding resistance. Assuming Mohr-Coulomb materials, it has been shown that the failure plane with a dip $\theta_c$ bisects the local topographic slope $\beta$, and the





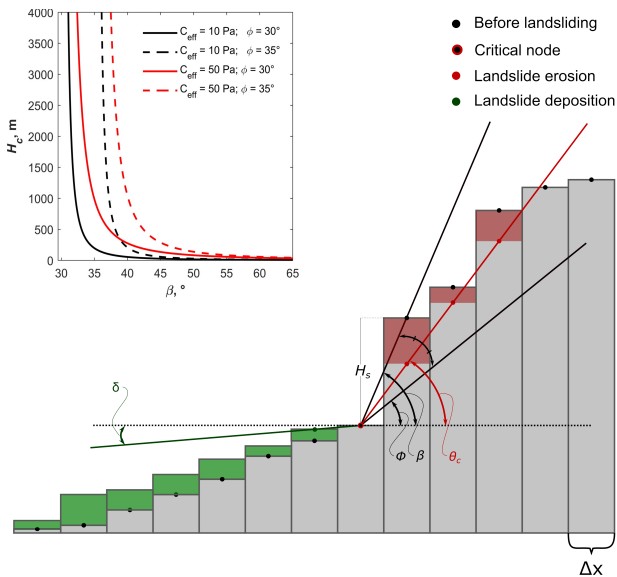

**Figure 3. Sketch of landslide algorithm in two dimensions.** Landslide erosion (red shaded area) is calculated using the Culmann approach (Culmann, 1875). $\beta$ is the topographic slope, $\phi$ represents the angle of internal friction and $\theta$ is the inclination of the rupture plane. Deposition of landslide-derived sediment (green shaded area) is calculated using a non-local diffusion equation (Eq. 11, cfr. Carretier et al., 2016). $\delta$ is the minimal angle of the spreading slope under which landslide-derived sediment is distributed on the hillslope. This sketch illustrates a case where none of the landslide-derived sediment is in permanent suspension ($F_{f_{hill}} = 0$) and the amount of eroded sediment (red shaded area) equals the amount of deposited sediment (green shaded area). If the deposited volume creates a down-slope gradient which is lower than the minimum spreading angle, $\delta$, the slope of deposited volume is adjusted so that the spreading slope equals $\delta$. Probability for sliding is calculated as the ratio of the local hillslope height $H_s$ to the maximum hillslope height $H_c$ (Eq. 7, cfr. Densmore et al., 1998). The inset plot illustrates that $H_c$ depends on the rock strength (cohesion $C$ and internal friction angle $\phi$) and the topographic slope $\beta$. The plotted lines are calculated using Eq. 8 with $\rho$ is 2700 $\mathrm{kg\,m^{-3}}$, and $g$ = 9.81 $\mathrm{m\,s^{-2}}$.

material's angle of internal friction $\phi$ (Densmore et al., 1998; Champel, 2002):

$$\theta_c = \frac{\beta + \phi}{2}. \tag{6}$$

The implementation of the Culmann theory in HyLands is illustrated in Fig. 3. For all points within the landscape where a
landslide is initialized, the failure plane dipping at $\theta_c$ is extended until it daylights (i.e., intersects the topographic surface).

Modeling andslide frequency and location depends critically on the identification of points in the landscape where landslides initiate. A wide variety of events ranging from co-seismic activity and peak ground acceleration (Meunier et al., 2007) over intense storm events (Marc et al., 2018) to human hillslope destabilisation (Guns and Vanacker, 2014) may trigger mass wasting
through landslide activity. Although these triggers could be added, HyLands mainly aims to simulate the impact of topographic landscape configuration on landslide activity. Therefore, we follow Densmore et al. (1998) in identifying unstable grid nodes



as points in the landscape where the topographic slope ($\beta$) exceeds the angle of internal friction ($\phi$). For all unstable nodes, the probability for sliding, $p_{LS}$, is calculated as:

$$p_{LS} = \frac{H_s}{H_c}. \tag{7}$$

where $H_s$ is the local hillslope height calculated as difference between every pixel in the landscape and it's highest neighbour (Fig. 3) and $H_c$ is the maximum stable hillslope height which is calculated as (Densmore et al., 1998; Champel, 2002):

$$H_c = \frac{4C}{\rho\, g} \frac{sin\beta\, cos\phi}{1 - cos(\beta - \phi)}. \tag{8}$$

Here $C$ is the cohesion [$\mathrm{M\,L^{-1}\,T^{-2}}$], $\rho$ is the rock density [$\mathrm{M\,L^{-3}}$] set to 2700 $\mathrm{kg\,m^{-3}}$, and $g$ the gravitational acceleration ($g$ = 9.81 $\mathrm{ms^{-2}}$). To simulate the random nature of landslides, grid nodes where landslides initiate are selected using a stochastic sampling scheme:

$$rnd\, \frac{dt}{t_{LS}} \begin{cases} > p_{LS} & \text{No Landsliding} \\ < p_{LS} & \text{Landsliding, select as critical node} \end{cases}. \tag{9}$$

where $rnd$ is a random number between 0 and 1 and $t_{LS}$ is the return time for landslides with $t_{LS} >= dt$. Unstable nodes where a landslide is induced will further be referred to as critical nodes (Fig. 3). Every model iteration, Eq. 9 is updated for all nodes where the topographic gradient ($\beta$) exceeds the critical material friction angle ($\phi$). From Eqs. 8 and 9, it follows that the probability for sliding depends on the topographic slope, $\beta$, and inversely correlates with the angle of internal friction, $\phi$, and the cohesion of the material, $C$ (see inset of Fig. 3). Cohesion is a scale-dependent variable and parameter values covering several orders of magnitude have been reported (Sidle and Ochiai, 2006). Jeandet et al. (2019) inverted several landslide inventories and found effective cohesion values ranging between 10 - 35 kPa. These values are lower than geomechanical values representing large-scale rock strength (e.g. Densmore et al., 1998; Champel, 2002) which is attributed to the decrease in rock cohesion in the vicinity of faults following earthquakes (Gallen et al., 2015). In our experiments, we will use values for cohesion in the range reported by Jeandet et al. (2019), which represent effective cohesion following an earthquake or a storm.

The landslide return time $t_{LS}$ controls the absolute number of critical nodes where landslides are initiated. If $t_{LS}$ equals the timestep $dt$, the number of landslides per timestep is solely controlled by the ratio $H_s/H_c$. When $t_{LS} >> H_s/H_c$, however, the number of landslides per timestep is reduced (Eq. 9). While the $H_s/H_c$ ratio controls the topographic location of landsliding onset and thus the landslide characteristics (size and volume), the landslide return time, $t_{LS}$ controls the absolute number of landslides and therefore overall landslide erosion rates.

HyLands enables the simulation of landslides at every location in the landscape. Every iteration, landslides are induced at critical nodes sampled using the probabilistic approach outlined above (Eq. 9, Figs. 3 and 4). We propose a recursive approach to calculate the magnitude of a single landslide. For every critical node, we build a stack of unstable (= sliding) nodes. The stack is initialized by adding the critical node. Next, a recursive procedure is applied until the stack is empty. This procedure exists of the following steps: (i) Select the first node from the stack (thus starting with the critical node). (ii) Evaluate all up-slope neighbouring nodes. If the elevation of a neighbouring node exceeds the elevation of the sliding plain defined by $\theta_c$, the





node is identified as a sliding pixel and added the to stack of landslide pixels. (iii) Remove the first node from the stack. This procedure is repeated until the stack is empty. HyLands offers the possibility to set a maximum landslide area ($A_{LS_{max}}$). Once

this maximum is achieved, or when no more pixels are added to the stack of landslide pixels, the landslide area is defined. All pixels inside this area are eroded to the elevation of the sliding plane, thereby adjusting both $E_{s_{hill}}$ and $E_{r_{hill}}$ of Eq. 1 for all pixels involved.

### 2.2.2 Flux of landslide derived sediment

The spatial change in hillslope derived volumetric sediment flux $Q_{s_{Hill}}$ $\left[L^3/T\right]$ per unit width $w$ $[L]$ is written as:

$$\frac{\partial\left(Q_{s_{Hill}}/w\right)}{\partial x} = E_{s_{hill}} + \left(1 - F_{f_{hill}}\right) E_{r_{hill}} - D_{s_{hill}}. \tag{10}$$

where $F_{f_{hill}}$ is a unitless fraction of fine hillslope derived sediment. The factor $1 - F_{f_{hill}}$ $[-]$ represents the idea that some fraction of the hillslope derived sediment is instantaneously evacuated as dissolved or suspended sediment (Page et al., 1999; Hovius et al., 2000; Lin et al., 2008; Tenorio et al., 2018), and therefore should not be tracked as sediment. When $F_{f_{hill}} = 0$, the system is fully mass conservative and all sediments produced by landslide activity contribute to the sediment flux (Eq. 1).

### 2.2.3 Deposition of landslide material

In HyLands, landslides can initiate at any point in the landscape (Fig. 4). A steepest descent flow routing algorithm is known to be unrealistic for flow and sediment redistribution on hillslopes (Pelletier, 2010). Moreover, the use a constant spreading slope, as suggested by Densmore et al. (1998) does not take into account the topographic relief when redistributing landslide derived sediment. When using a constant sediment spreading slope, sediment deposited on flat parts of the landscape is spread

out over a much longer distances than sediment deposited on steep parts, where the the large difference between topographic slope and spreading slope can accommodate for large sediment volumes. This is not realistic as sediment travel distance should depend on topographic gradient: material traveling over steeper slopes should go farther, all else being equal (Roering et al., 1999; Campforts et al., 2016).

A common approach to simulate sediment transport and deposition on hillslopes while considering the topographic gradient,

is the use of linear or nonlinear diffusion equations (Roering et al., 1999; Andrews and Hanks, 1985). However, such approach is not suited to simulate the distribution of landslide-derived sediment. While diffusion equations distribute sediment only between the neighbouring cells of a pixel, landslide derived sediment has run-out distances which can be significantly longer than a single grid cell (Claessens et al., 2007). Therefore, we adopt a non-linear, non-local deposition scheme for landslide derived sediment outlined by Carretier et al. (2016):

$$D_{s_{hill}} = \frac{Q_{s_{Hill}}/w}{L} \tag{11}$$

where $D_{s_{hill}}$ $[L/T]$ is the volumetric deposition flux of hillslope derived sediment per unit bed area and $L$ [L] represents a sediment transport distance. The larger $L$, the bigger the distance over which sediments are transported and the lower the local





deposition rate. $L$ is calculated for every grid cell as:

$$L = \frac{dx}{1 - \left(\frac{S}{Sc}\right)^2} \qquad (12)$$

where $S_c$ is a critical slope, which we furhter assume to be equal to the angle of internal friction ($\phi$). When the hillslope gradient $S << S_c$, most of the incoming sediment will be deposited and the resulting outcome is similar to the one obtained using a regular diffusion equation, also referred to as a local solution (Furbish and Roering, 2013; Carretier et al., 2016). When $S$ approaches $S_c$, $L$ goes to infinity implying that no deposition will occur at the considered cell. At steep slopes, sediment transport therefore shows non-local behaviour in the sense that erosion activity of non-local, upstream cells is integrated when calculating the sediment flux $Q_s$ (Carretier et al., 2016). When $S > S_c$, $L$ is set to $inf$. For negative values of $S$, which might occur for flooded nodes, $L$ is set to $dx$. In HyLands, a minimal spreading angle ($\delta$) can still be imposed under which landslide derived sediments are deposited but is not required (Fig. 3).

Contrary to fluvial dynamics (SPACE) where a single flow direction algorithm (steepest descent) is used, landslide-derived sediment is spread over the hillslopes using a multiple flow direction algorithm redistributing sediment over the downstream cells in proportion to the local slope (Fig. 4, cfr. Carretier et al., 2016). When a lot of sediment is debouched into fluvial channels, rivers can be blocked by landslide dams. HyLands uses a lake identification algorithm to identify flooded nodes during every model iteration. Lakes are identified by filling all sinks in a landscape to the brim. By default, flooded nodes do not erode but do allow for sediment deposition (Eq. 5).

In the remainder of this paper, we will first evaluate the performance of the fluvial and landslide components of HyLands through a set of verification and validation runs. Next, the coupling between landslide activity and long term landscape evolution will be evaluated using a synthetic model setup where a steady-state landscape is exposed to a pulse of landslides. All model experiments executed in the framework of this paper are available as executable Matlab scripts and as dynamic landscape evolution movies (Table 2).

## 3 Verification and evaluation

### 3.1 Comparison to analytical solutions for the fluvial dynamics

In the first three test runs (detachment, transport-limited and mixed), a steady-state artificial landscape is simulated using a square grid of 20 by 20 nodes with a coarse spatial resolution of 100 m. The run is initialized from a surface with randomly generated microtopography. The initial surface is a tilted plain which drains towards the southwestern corner, the only open boundary node. Therefore sediment and water can only leave the domain through this southwestern corner. The setup is identical to the one proposed by Shobe et al. (2017) in order to facilitate comparison. The timestep is set to 10 years. Under detachment-limited conditions (imposed by setting $F_f = 1$), the sediment thickness $H$ equals 0 everywhere and through the entire model run and sediment produced by river incision into bedrock is instantaneously evacuated from the simulated domain. When assuming that water discharge is proportional to the drainage area ($q \propto A^m$) it has been shown that under steady-state





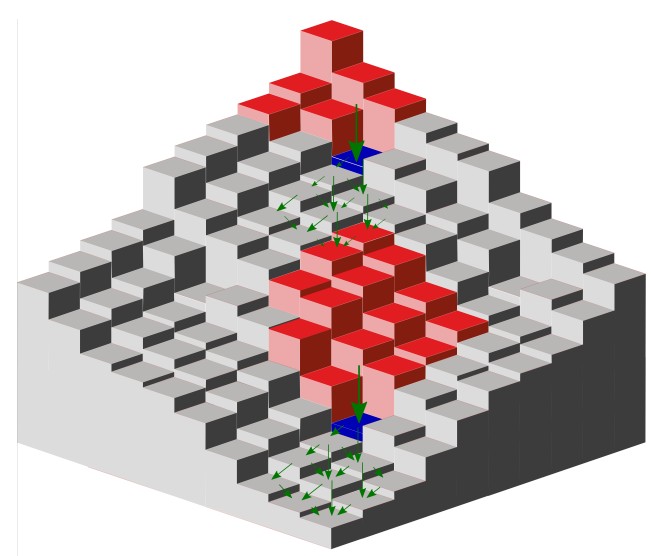

**Figure 4. Sketch of landslide algorithm in three dimensions.** Cells shaded in blue indicate the critical nodes where landslides initiate. Cells shaded in red represent the landslide source areas. After mass failing, sediment will be redistributed over the downslope cells using a multiple flow direction algorithm (indicated with green arrows). Sediment deposition rate depends on a transport distance $L$ (cfr. Eqs. 11 and 12).

conditions, fluvial erosion results in the following slope-area relationship (Shobe et al., 2017):

$$S = \left( \frac{U}{K_r A^m} \right)^{1/n} \tag{13}$$

Fig. 5.b illustrates that when using parameter values listed in Table 1, HyLands reproduces the slope-area relationship given by Eq. 13. Similarly, it can be shown that under transport-limited configurations where $H >> H_*$, the theoretical slope-area relationship for fluvial incision can be written as (Shobe et al., 2017):

$$S = \left[ \frac{V}{r} + 1 \right]^{1/n} \left[ \frac{U}{K_s} \right]^{1/n} A^{-m/n} \tag{14}$$

where $r$ represents the runoff rate $[L/T]$. To mimic a transport limited configuration, we run HyLands assigning an initial sediment thickness $H$ of 100 m. Other parameters values are shown in Table 1. Figure 5.d illustrates the slope-area plot for all nodes of the simulated steady-state landscape showing a close match with the analytical prediction (14). Moreover, HyLands also reproduces the theoretical steady-state sediment flux relationship for transport-limited conditions (Shobe et al., 2017):

$$Q_{S_{fluv}} = U A. \tag{15}$$

Finally, we evaluate the hybrid nature of the SPACE component in simultaneously simulating fluvial bedrock incision and sediment dynamics. Under such configuration the slope-area relationship can be written as (Shobe et al., 2017):

$$S = \left[ \frac{UV}{K_s A^m r} + \frac{U}{K_r A^m} \right]^{1/n} \tag{16}$$





**Table 1.** Parameter values

| | No Landsliding | | | Landsliding | | | |
| --- | --- | --- | --- | --- | --- | --- | --- |
| | Detachment-limited | Transport-limited | Mixed | Namche-Barwa | \multicolumn{3}{c}{Synthetic(a)} | | |
| | | | | | Pre | LS-Event | Post |
| Number of rows $(-)$ | 20 | 20 | 20 | 1918 | | 75 | |
| Number of columns $(-)$ | 20 | 20 | 20 | 1149 | | 75 | |
| Node spacing (m) | 100 | 100 | 100 | 20 | | 20 | |
| Time step (yr) | 10 | 10 | 10 | 5 | | 5 | |
| Run time (kyr) | 100 | 100 | 200 | 500 | $5\times10^{6}$ | 100 | $5\times10^{4}$ |
| Initial $H$ (m) | 0 | 100 | 0 | 0 | 0 | varying | varying |
| $U$ (myr$^{-1}$) | $1\times10^{-4}$ | $1\times10^{-4}$ | $1\times10^{-4}$ | 0 | | $1\times10^{-3}$ | |
| $K_r$ (m$^{-1}$) | $1\times10^{-3}$ | $1\times10^{-4}$ | $5\times10^{-3}$ | $5\times10^{-4}$ | | $5\times10^{-5}$ | |
| $K_s$ (m$^{-1}$) | 0.01 | 0.01 | 0.01 | $1\times10^{-3}$ | | $7.5\times10^{-5}$ | |
| $m$ $(-)$ | 0.5 | 0.5 | 0.5 | 0.5 | | 0.5 | |
| $n$ $(-)$ | 1 | 1 | 1 | 1 | | 1 | |
| $H_*$ (m) | 1 | 1 | 1 | 2 | | .5 | |
| $\phi_{sed}$ $(-)$ | 0 | 0 | 0 | 0 | | 0 | |
| $F_{f_{fluv}}$ $(-)$ | 1 | 0 | 0 | 0 | | 0 | |
| $V^{(b)}$ (myr$^{-1}$) | 1 | 5 | 5 | 2 | | 2 | |
| $V_{Lake}^{(b)}$ (myr$^{-1}$) | 1 | 5 | 5 | 10 | | 10 | |
| $C$ (kPa) | - | - | - | 15 | - | 15 | - |
| $\phi$ (°) | - | - | - | 38 | - | 35 | - |
| $t_{LS}$ (yr) | - | - | - | $2\times10^{4}$ | - | $2\times10^{3}$ | - |
| $\delta$ (°) | - | - | -0.01 | - | 0.01 | - | |
| $F_{f_{hill}}$ $(-)$ | 1 | 0 | 0 | 0.25 | - | 0.25 | - |

Not all parameters will influence the model outcome in all cases. For example, the value of $V$ is irrelevant for the detachment-limited case when all eroded bedrock passes out of the model domain as permanently suspended fine sediment ($F_{f_{fluv}} = 1$). Landslide parameters are only relevant for models where landslide activity is simulated.

$^{(a)}$: The synthetic landscape evolution model consists of three consequent stages, a pre-landslide stage, a landslide stage and a post-landslide stage. Only parameter values which differ for these stages are listed in the table

$^{(b)}$: HyLands enables spatially variable values for $V$ (Eq. 5) to distinguish between settling velocities over flooded versus non-flooded nodes by changing the values for respectively $V$ and $V_{Lake}$

Under steady-state, both the height of the bedrock and the sediment layer should remain unchanged so that:

$$\frac{\partial \eta}{\partial t} = \frac{\partial R}{\partial t} + \frac{\partial H}{\partial t} = 0 + 0 = 0 \qquad (17)$$

which leads to a constant soil thickness over the landscape given by (Shobe et al., 2017):

$$H = -H_* ln \left[ 1 - \frac{V}{\frac{K_s r}{K_r} + V} \right]. \qquad (18)$$




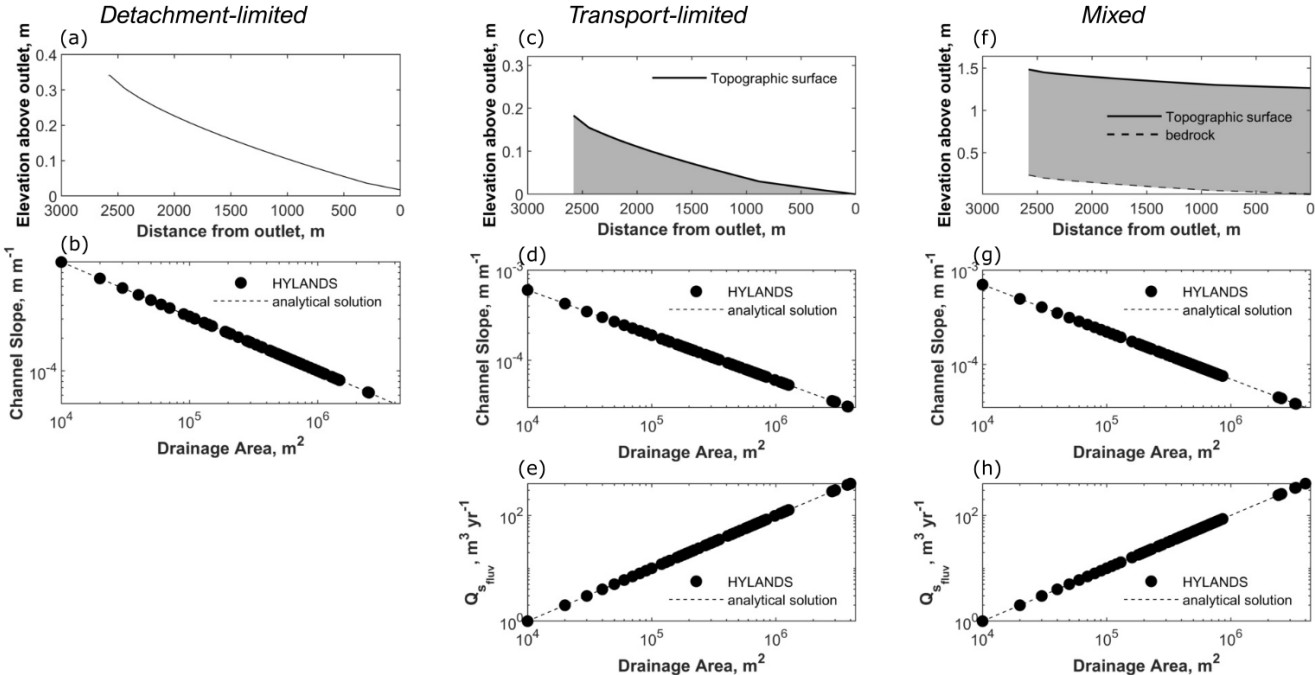

**Figure 5. Verification of fluvial SPACE component**. **(a)** Longitudinal profile of the trunk stream at steady state simulated under detachment-limited conditions where no sediment is present because $F_{f_{fluv}} = 1$ and all produced sediment is assumed to be evacuated instantaneously. At steady state, the profile evolves towards a concave-upward profile, being in equilibrium with the imposed uplift which is reflected in the steady-state slope-area relationship **(b)** matching the predicted analytical solution (Eq. 13). **(c)** Longitudinal profile of the trunk stream at steady state simulated under transport-limited conditions. At steady state, the profile evolves towards a concave-upward profile, being in equilibrium with the imposed uplift which is reflected in the steady-state slope-area relationship **(d)** matching the predicted analytical solution (Eq. 14). **(e)** illustrates the steady-state fluvial sediment flux $(Q_{s_{fluv}})$ as a function of the drainage area and matches with the predicted analytical flux-area relationship (Eq. 15). **(f)** Longitudinal profile of the trunk stream at steady state simulated under mixed alluvial-bedrock conditions. At steady state, both the topographic and bedrock profiles are in equilibrium with the imposed uplift which is reflected in the steady-state slope-area relationship **(g)** matching the predicted analytical solution (Eq. 16). **(h)** illustrates the steady-state fluvial sediment flux $(Q_{s_{fluv}})$ as a function of the drainage area and matches with the predicted analytical flux-area relationship under hybrid sediment-bedrock fluvial incision dynamics (Eq. 15).

To evaluate the performance of the hybrid fluvial dynamics, we run HyLands to a steady-state, starting from an initial surface without any sediment cover and using parameter values listed in Table 1. The obtained slope-area relationship (Fig. 2.g) matches with the theoretical relationship Eq. 16, so does the soil thickness $H$ which evolves toward a constant thickness (Eq. 18) and the sediment flux relationship honoring (Eq. 15).

290





## 3.2 Evaluation of the landslide component

Because landslide activity is a stochastic process, it not possible to derive an exact, analytical, solution to evaluate the performance of the landslide component in HyLands. However, it has been shown that most landslide inventories obey consistent magnitude-frequency and magnitude-volume relationships (Malamud and Turcotte, 1999; Stark and Hovius, 2001; Guzzetti et al., 2002; Korup, 2005; Guns and Vanacker, 2014; Larsen and Montgomery, 2012). To evaluate the performance of HyLands, we run the model over a limited amount of time for an area where both relationships are well constrained. The performance of the landslide module is evaluated based on its capacity to reproduce those calibrated relationships.

### 3.2.1 Landslide scaling relationships

A first empirical universal relationship is the landslide magnitude-frequency distribution that describes the number of landslide events of a given size. This relationship is characterized by a negative power law for landslides having an area greater than a given threshold value and a characteristic rollover for smaller landslides (Stark and Hovius, 2001; Malamud and Turcotte, 1999; Guzzetti et al., 2002). Magnitude-frequency distributions are typically described using a three parameter inverse gamma distribution as (Malamud et al., 2004):

$$p(A_L; \rho_l, a_l, s_l) = \frac{1}{a_l * \Gamma(\rho_l)} \left[ \frac{a_l}{A_L - s_l} \right]^{\rho_l + 1} exp \left[ \frac{a_l}{A_L - s_l} \right] \tag{19}$$

where $A_L$ is the landslide area $\left[ L^2 \right]$, $p(A_L)$ is the probability density of a landslide area $(A_L)$, $a_l$, $s_l$ and $\rho_l$ are empirical parameters, and $\Gamma(\rho_l)$ is the gamma function of $\rho_l$. A second empirical universal relationship, is the volume-area scaling relationship where the volume $V_L$ of a given landslide is a function of its area $A_L$ as (Hovius et al., 1997):

$$V_L = \alpha_l A_L^{\gamma_l} \tag{20}$$

where $\alpha_l$ is an intercept and $\gamma_l$ a scaling exponent.

### 3.2.2 Applying HyLands to the Namche Barwa-Gyala Peri massif

To evaluate to the performance of HyLands, the model is applied to a digital elevation model (DEM) of the Eastern Himalaya where the Yarlung Tsangpo river cuts through the Namche Barwa-Gyala Peri massif (Fig. 6). The area is characterized by rapid exhumation (King et al., 2016), steep topography, and steep river gradients causing high stream power (Finnegan et al., 2008). To quantify erosion rates in the area, Larsen and Montgomery (2012) mapped more than 15,000 landslides and constructed an inventory of landslides pre-dating 1974 and an inventory containing all landslide events between 1974 and 2007 (Fig 8.a). We use this area solely to demonstrate and evaluate the performance of HyLands and do not aim to reproduce exact features of landscape exhumation in this region.

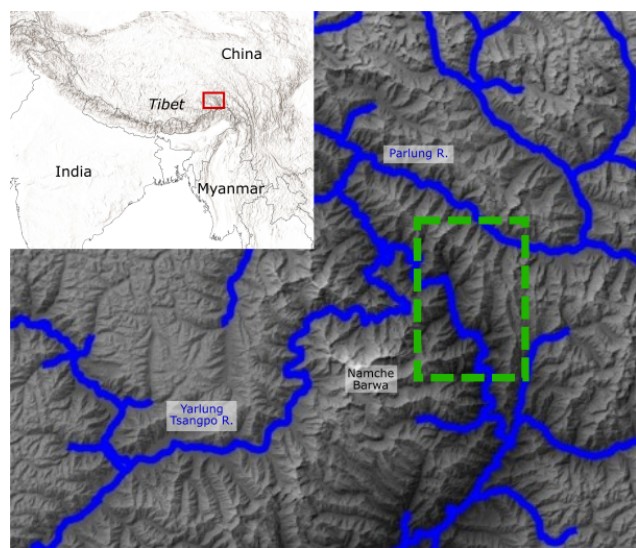

**Figure 6. Namche Barwa-Gyala Peri massif used for model evaluation**. The red rectangle on the inset figure indicates the geographical location of the study area. The green dashed rectangle indicates the part of the DEM used to evaluate HyLands. The shaded colours indicate elevations, which were derived from the 30 m SRTM v3 DEM (NASA JPL, 2013) and resampled to a higher resolution of 20 m using a bicubic interpolation method. Main map is produced with TopoToolbox (Schwanghart and Scherler, 2014). Inset map is made in QGis 3©, using Natural Earth vector and raster map data available at www.naturalearthdata.com.

### 3.2.3 Model parameterization

We run HyLands using the NASA Shuttle Radar Topography Mission (SRTM) v3.0 elevation data as an initial surface (Farr
et al., 2007), resampled to a higher resolution of 20 m using a bicubic interpolation method. As shown in Fig. 6, we only simulate part of the larger Namche Barwa-Gyala Peri massif studied by (Larsen and Montgomery, 2012). We selected a region mostly free of glaciers and surrounding the section of the Yarlung Tsangpo river where unit stream power is very high (ranging between 500 - 4000 W.m$^2$, (Finnegan et al., 2008). The simulated grid is composed of 1918 × 1149 nodes, covering a total area of ca. 960 km$^2$. We simulate landscape evolution over 500 years, using time steps of 5 years. For this experimental run,
we assume that there is no uplift. We acknowledge that this condition is not met in the area, but imposing an uplift field is not necessary for evaluating the performance of the HyLands landsliding algorithm. Inserting realistic uplift patterns to simulate the dynamic evolution of the area would require (i) implementation of the complex tectonic configuration of the area (King et al., 2016) and (ii) simulation of a bigger area to capture the dynamic interplay between uplift and river dynamics. This is beyond the scope of the application and we therefore assume that the tectonic configuration controlling landscape evolution
over the limited timescale simulated in this experiment (500 years) is captured by the topography of the area (Kirby and Whipple, 2012).

We run HyLands assuming open boundary conditions: all sediment produced within the domain through river incision and landsliding can be exported from the domain across any of the four boundaries. For simplicity, we use a simple stream power





**Table 2.** Simulation movies and script names

|  |  |  | Script name on GitHub | Link to movie |
|---|---|---|---|---|
| No Landsliding | Detachment-limited | | HyLands-NoLS-DL.m | https://doi.org/10.5446/45969 |
| | Transport-limited | | HyLands-NoLS-TL.m | https://doi.org/10.5446/45967 |
| | Mixed | | HyLands-NoLS-Mixed.m | https://doi.org/10.5446/45968 |
| Landsliding | Real DEM, Namche-Barwa | | HyLands-LS-NB.m | https://doi.org/10.5446/45973 |
| | Synthetic | Before intense LS period | HyLands-LS-B-LS.m | https://doi.org/10.5446/45970 |
| | | intense LS period | HyLands-LS-LS.m | https://doi.org/10.5446/45971 |
| | | After LS period | HyLands-LS-A-LS.m | https://doi.org/10.5446/45972 |

formulation for river incision where thresholds for both sediment entrainment and bedrock erosion are negligible. Standard

scaling exponents are used ($m = 0.5$ and $n = 1$ in Eq. 4) and the bed sediment porosity and the fraction of fine river sediments are assumed to be zero ($\phi_{sed} = 0$ and $F_{f_{fluv}} = 0$ in Eq. 1). We calculate landslide activity using the landslide module of HyLands and assume that 25% of the landslide-derived sediment is evacuated out of the system as fine material ($F_{f_{hill}} = 0.25$). We assume that the angle of internal friction ($\phi$) is comparable to the mode of the topographical slope distribution (Burbank et al., 1996; Korup, 2008; Montgomery and Gran, 2001), reported to range between 37°- 39° and here set to 38° (Larsen and

Montgomery, 2012). Cohesion $C$ is known to vary over a wide range and strongly depends on rock mechanical properties (Wyllie and Mah, 2017). Site-specific calibration would require detailed mapping of lithologcal units and we therefore set $C$ to 15 kPa, a value in the range of previously optimized cohesion for the Himalaya (Jeandet et al., 2019, 12-20Pa in ). Cohesion and the angle of internal friction influences the size-distribution of landslides in several ways (Jeandet et al., 2019). The angle of internal friction $\phi$ controls the angle of the potential rupture plane such that lower values of $\phi$ will result in lower rupture

dipping angles, which, for the same topographical configuration, results in thicker and larger landslides. The effective rock cohesion value $C$ influences the critical hillslope height $H_c$ in Eq. 6. Larger values values for $C$ will result in larger values for $H_c$, thus decreasing the probability of landslides on less steep slope sections and resulting in fewer small landslides. The minimum value of the spreading slope under which landslide-derived sediment is redistributed on hillslopes is set to 0.01°. We set the return time for landsliding $t_{LS}$ to $2 \times 10^4$ years. $t_{LS}$ regulates the probability that unstable cells evolve into a landslide

(Eq. 9) and therefore controls the number of landslide events per timestep $\Delta t$. When applying HyLands to reconstruct or predict landslide activity, $t_{LS}$ should be a function of the frequency (or return time) of triggering events (large earthquakes or rainfall events). A full overview of the model parameters is given in Table 1.

### 3.2.4 Model evaluation results

Figure 7 shows time slices of the model run after 5 (initial iteration), 165, 330 and 500 (final iteration) model years. Locations

for landslide initiation (critical nodes) are well spread over the landscape. The number of landslide events (the number of

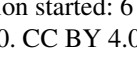



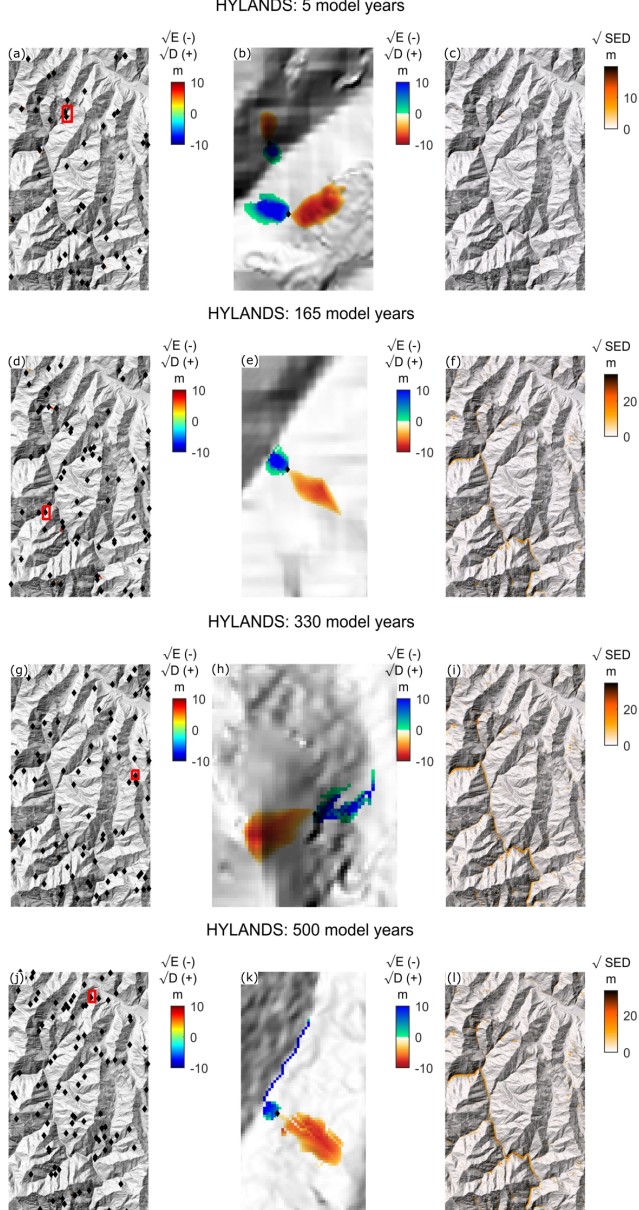

**Figure 7. Timeslices of HyLands model run for the Namche Barwa region after 5, 165, 330 and 500 model years.**(**a, d, g, j**) Indicate the location of the landslides at the given timestep (black dots). The colors represent the square root (used for ease of visualization) of the landslide erosion (-) and deposition (+) during the given time step. (**b, e, h, k**) Zoom of simulated landslide erosion/deposition indicated with the red square in respectively (a, d, g and j). (**c, f, i, l**) Represent the square root of the total amount of sediment, generated through river incision (SPACE module) and landsliding after respectively 5, 500, 1500 and 2000 model years. The grey shaded colours indicate elevations, which were derived from the 30 m SRTM v3 DEM (NASA JPL, 2013).





black dots in Fig. 7) is mainly controlled by the return time for landsliding $t_{LS}$. The fan-shaped deposition zones of landslide-derived sediments reflect the use of a multiple flow sediment routing algorithm. Landslide-derived sediment predominantly accumulates at hillslope toes, as well as in or near river channels. Some accumulation also occurs on hillslopes. Overall, landslides strongly influence the thickness of the alluvial bed sediment layer. Note that the shape of the erosion and deposition
zones adjust through the course of the model run. The presence of previous landslide activity alters the topographic relief and hence determines which pixels become susceptible to erosion and deposition as landscape evolution continues (Fig. 7).

To quantitatively evaluate the performance of the HyLands landslide module, we compare modelled landslide properties against observed scaling relationships (Eq. 19 and 20). Figure 8.a compares the modelled and observed magnitude-frequency distribution. We observe good correspondence between the model and the data, with the power-law tail of the distribution
falling within the envelope defined by the two inventories of Larsen and Montgomery (2012). Similarly to observed magnitude-frequency distributions, HyLands simulates the rollover or the transition from an increasing magnitude-frequency relationship to a decreasing one. This observation confirms that the shape of landslide magnitude-frequency distributions can be explained using mechanical landslide processes (cfr. 2.2.1) and the geometry of the studied region (Jeandet et al., 2019). Figure 8.b shows that HyLands is capable of approaching the universal Area-Volume relationships found by Larsen et al. (2010). While HyLands
seems to overestimate simulated landslide volumes for very small landslides, the fit between HyLands and the observed relationship improves for large landslides. We attribute overestimation of landslide volumes for smaller landslides to the nature of the landslide algorithm: while HyLands does not allow any sediment to be deposited within the landslide scar, this typically does occur in nature. Future developments of the algorithm could allow in-scar deposition for more realistic simulations. Moreover, there is a resolution effect: due to DEM noise or heterogeneity, the algorithm might select small landslides of one or
two cells on very steep hillslope patches thus resulting in high landslide volumes. However, such steep hillslope patches might represent noise in the DEM rather than actual steep slopes. The use of high resolution DEMs could partly resolve this issue.

### 3.3    Model application

The explicit coupling of landslides and landslide-derived sediment to long-term landscape evolution enables the study of a wide range of interactions which otherwise can only be inferred or partially simulated. A common application is to evaluate
the coupling between landslides and river-bed morphology. To evaluate the impact of a landslide event on long-term channel profile evolution, we run a synthetic landscape evolution model to steady state. After 5 million years, we simulate a period of 100 years with intense landslide activity, analogous to a period of elevated landslide activity triggered by a series of seismic events. After 100 years of landslide activity, we assume landslides are no longer triggered and let the landscape evolve back to its original steady state. Such an experiment not only allows evaluation of the extent to which landslides perturb the topography
of river profiles, but also enables estimates of the time required for a landscape to respond to a major perturbation (e.g. a series of earthquake-triggered landslides).

The model run consists of three stages. In the first stage, the model is run to a steady state. For reasons of comparability, we simulate landscape evolution on a grid similar to the one used for the verification runs (section 3), i.e. with a single open boundary node in the southwestern corner. To simulate more realistic landscape scales, we use a domain of 75 by 75 nodes



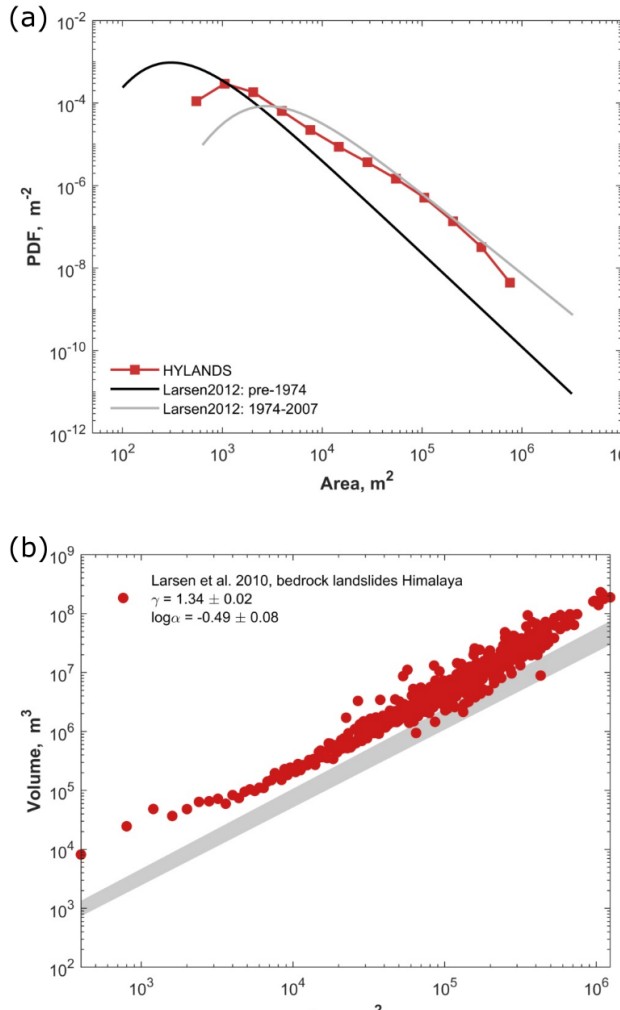

**Figure 8. Comparison between modelled and observed characteristic landslide scaling relationships.** (a) Magnitude frequency relationship. The grey and black line represent the best fitting inverse gamma distribution (Eq. 19) of the landslide activity mapped before 1974 and between 1974 - 2012, respectively (Larsen and Montgomery, 2012). The fitting parameters are respectively ($a_l = 768 \times 10^{-6}$, $s_l = -32.6 \times 10^{-6}$ and $\rho_l = 1.27$) and ($a_l = 6100 \times 10^{-6}$, $s_l = -311 \times 10^{-6}$ and $\rho_l = 0.96$)). The red dots represent the magnitude-frequency distribution simulated using HyLands. (b) Area-Volume relationship. The grey dashed zone represents the expected area-volume scaling relationship, observed for bedrock landslides in the Himalaya (Larsen et al., 2010). Fit is calculated using Eq. 20 with fitting parameters $\gamma = 1.32 \pm 0.02$ and $^{10}\log\alpha = -0.49 \pm 0.06$. The red dots represent the geometry of the data simulated with HyLands.

with a higher resolution of 20 m. A complete overview of model parameter values is given in Table 1. The evolution of the landscape over time is shown in a series of time slices (Fig. 9.a-c). Model behaviour at this stage is as expected for the SPACE river erosion model when hillslope processes are not explicitly simulated (Shobe et al., 2017). During the first timesteps, the






**Figure 9. Synthetic model run showing landscape evolution to steady state followed by an intense landsliding period of 100 years.**
(a-d) Time slices showing evolution of the landscape to steady state, before the landslide period. (e-h) Time slices showing the landslide period where intense landsliding is occurring over a period of 100 years. Note that, during landsliding, both pure landslide dams arise as well as irregularities in the bedrock profile (the grey bumps). The latter originate from the river being redirected after landsliding forming epigenetic gorges (see text).





drainage network establishes and the landscape gradually approaches a steady state with uniform sediment thickness across the entire landscape.

In the second stage (Fig. 9.d-f), we simulate a period of intense landslide activity by triggering a large number of landslides. Landslides are initiated based on their probability of sliding (Eq. 7) assuming an internal friction angle of $35°$ and a low landslide return time ($t_{LS} = 2 \times 10^3$ years). Under this configuration, many of the steep portions of the landscape become prone to landslide erosion and transform into a landslide source area. We assume that 25% of the landslide-derived sediment is instantaneously evacuated out of the system as fine material ($F_{f_{hill}} = 0.25$). Landslides trigger the formation of landslide

dams, resulting in flooded river sections. Landslide dams not only alter the topographic elevation of the simulated domain but also change the drainage network. The location of the river bed can change due to landslides and landslide-derived sediment rearranging the valley-bottom topography. This is why the bedrock profile of the plots shown in Fig. 9.e-h has a bumpy shape at several locations along the profile.

Immediately after the intense landsliding period, the trunk stream of the drainage network is choked with sediment and landslide dams are abundant (Fig. 10.a-f). In the first few thousand years following the intense landsliding period, the lakes

gradually fill in with sediment. After 1500 years, most of the landslide-dammed lakes are filled with sediment. The fluvial profile is now characterized by a chain of knickpoints characteristic for fluvial profiles experiencing the delivery of immobile debris by landslides (Ouimet et al., 2007) or other hillslope processes (Shobe et al., 2016, 2018). Where the in-channel bedrock is not covered with sediment, river incision into bedrock continues. However, upstream of landslide dams, the bed is choked

with sediment and the alluvial cover is too thick for bedrock incision to continue. As bedrock uplift continues and sediment is slowly been evacuated from from filled lakes, the bedrock profile adjusts and small knickpoints are created along the river profile. While the specific cause is different (landslide dams ponding sediment vs. delivery of large-grained colluvium), the mechanism of knickpoint generation is similar to the numerical simulations of Shobe et al. (2016) and Shobe et al. (2018) in that a bare-bedrock reach downstream of a sediment-mantled reach can undergo faster erosion, thereby generating knickpoints

that are decoupled from the baselevel signal.

There are two distinct mechanisms for the generation of irregularities in the channel profiles: drainage re-routing due to landslide dams and knickpoint generation due to spatially varying sediment cover triggering differential erosion. The former mechanism can result in reaches where the bedrock slope is adverse relative to the water surface slope (the bumps in the bedrock profile in Fig. s 9 and 10). The latter creates variability in the magnitude, but not the direction, of bedrock slope. The

drainage re-routing mechanism dominates in the simulations presented here (Fig. 10).

## 4    Discussion

Landscapes are the outcome of external perturbations, such as climate or tectonic variability, and internal dynamics originating from the coupling between fluvial incision and hillslope response (Burbank and Anderson, 2011; Glade et al., 2019). Much effort has been devoted towards understanding the relationship between fluvial erosion efficiency and climate variability both

through theoretical developments (Tucker, 2004; Lague, 2014) and observations (DiBiase and Whipple, 2011; Ferrier et al.,



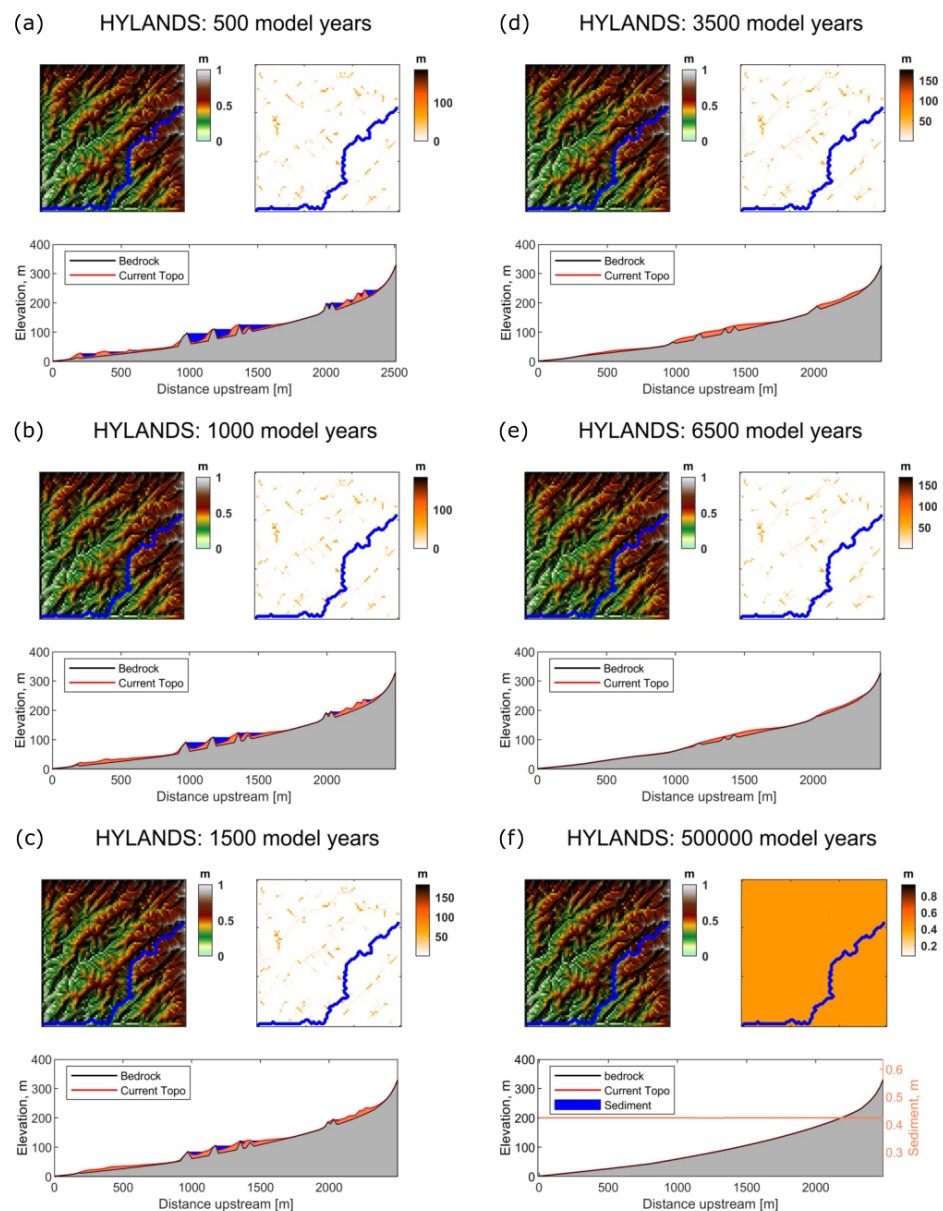

**Figure 10. Synthetic model run showing recovery from intense landsliding illustrated in Fig. 9**. **(a-h)** Time slices showing reestablishment of the landscape steady state. Bedrock bumps created by landslide-induced drainage redirection are eroded and the channel re-attains its smoothly concave-up, steady-state configuration.





2013). A main finding of those authors is that the role of allogenic fluvial response (i.e. transient adjustment to an external perturbation) can only be understood when considering autogenic fluvial dynamics such as the existence of incision thresholds (Snyder et al., 2003; Lague et al., 2005) and the internal lithological heterogeneity in a landscape (Campforts et al., 2019; Glade et al., 2019). However, the role of landslides in long-term landscape evolution, and especially the dynamic interaction

between river incision and landslide activity, is only poorly understood. HyLands offers a tool to study dynamic feedbacks between landslides and river incision. The role of sediment dynamics in altering fluvial erosion and sediment transport is clearly illustrated in the numerical experiment (Fig. 9) where 5-10 kyr are required for the landscape to evolve back to a steady sate after a pulse of landsliding. Not only does the delivery of landslide-derived sediment to the channel bed alter the topography of the channel profile, it also results in the formation of bedrock knickpoints and associated retreating incision

waves. HyLands thereby corroborate earlier observations that landslides, and tight channel-hillslope couplings in general, are autogenic mechanisms altering the way in which landscapes respond to external (allogenic) perturbations (Ouimet et al., 2007; Shobe et al., 2016; Glade et al., 2019). HyLands is designed to study the dynamic feedbacks between landslides and river erosion at large spatial and temporal scales. To do so, the model integrates an algorithm for deep-seated landsliding with a recently proposed model for fluvial incision (Shobe et al., 2017). HyLands enables simulations over several millions of years

and reproduces analytical predictions for fluvial dynamics and observed landslide scaling relationships.

## 4.1 Fluvial component

The SPACE river erosion model, which governs river evolution in HyLands, advances on existing river incision models in that it explicitly simulates the role of sediment in reducing the efficiency of bedrock incision (Beaumont et al., 1992; Lague, 2010). However, like SPACE, HyLands does not simulate the effect of increased bedrock incision efficiency due to mobile sediment

acting as eroding tools (Sklar and Dietrich, 2004). Field observations warrant consideration of the tool effect (Cook et al., 2013; Beer et al., 2017), and theoretical predictions have shown that the interaction between sediment and bedrock incision is adjusted when the tool effect is considered (Gasparini et al., 2007). The impact of explicitly simulating the tool effect due to landslide-derived sediment has been evaluated in a numerical modelling study (Egholm et al., 2013). Egholm et al. (2013) concluded that landslide activity and its delivery of abrasive agents to the channel accelerate fluvial incision in actively uplifting

mountain regions, whereas the lack of landslides in tectonically inactive mountain ranges strongly decreases erosion efficiency and enables topographic preservation. Adding the tool effect and evaluating its potential importance is therefore a primary goal for further model development in HyLands. Simulating the tool effect of sediments can be achieved by making the bedrock erosion function dependent (Eq. 4) on the sediment flux $Q_s$ (cfr. e.g. Gasparini et al., 2007; Hobley et al., 2011).

Like SPACE, HyLands does not include process-based approaches (Kean and Smith, 2004; Wobus et al., 2006; Davy and

Lague, 2009; Coulthard et al., 2013), simplified width adjustment rules (Lague, 2010; Yanites, 2018), or empirical closures (Attal et al., 2008) to dynamically calculate river width adjustments though time. HyLands assumes a relationship between drainage area and river width depending on the scaling exponent $m$ (fixed in our simulations to 0.5; see also Table 1). This approach implies that all river cells in the landscape are assumed to occupy 1 grid cell with distance $dx$, that channel width may be less than, equal to, or greater than $dx$, and that river width is only a function of contributing drainage area. It has





however been shown that river width might vary as a function of sediment flux or under varying tectonic configurations (Amos and Burbank, 2007; Turowski et al., 2009). Recent work using a 2D hydro-sedimentary numerical model (Davy et al., 2017) based on the Saint-Venant Equations has shown that river re-organisation and narrowing after landslide events might strongly increase sediment transport capacity and alter sediment evacuation time after big landslide events (Croissant et al., 2017). While simulating dynamic river width reorganisation at the landscape scale is currently not possible over longer timescales

due to computational limitations, generic approximations for the landslide triggered channel narrowing (Croissant et al., 2019) could be integrated in future versions of HyLands.

## 4.2   Landslides

The landslide algorithm in HyLands is based on finite slope mechanics and assumes a planar rupture plane geometry. Although our approach reproduces observed magnitude-frequency and area-volume scaling relationships and is supported by previous

work where landslides have been simulated using planar rupture planes (Jeandet et al., 2019), the use of more advanced rupture plane geometries has been proposed. Gallen and Wegmann (2017) for example propose the use of concave-upward rupture planes to simulate co-seismic landsliding. However, their approach is based on the statistical aggregation of one-dimensional slope-stability solutions and therefore does not fully honor the three-dimensional topographic complexity of the topographic surface like HyLands does. Evaluating the role of varying rupture plane geometries in three dimensions is one of the potential

future developments of HyLands.

At this stage HyLands does not explicitly simulate shallow landsliding which typically occurs at the interface between the bedrock and the overlying sediment/regolith cover. Given the existing ability of HyLands to simultaneously simulate bedrock evolution and sediment thickness, adding a shallow landslide algorithm is feasible (Montgomery and Dietrich, 1994; Claessens et al., 2007) and would further our understanding of the coupling between climate variability and landscape stability (Parker

et al., 2016). For shallow landslides to be added as a component in HyLands, the implementation and calibration of a regolith formation and soil flux model will however be required (Campforts et al., 2016). Using additional model components to simulate soil formation and transport requires calibration of several additional processes components; care is needed to prevent over-calibration and -parameterization of the model (Van Rompaey and Govers, 2002).

Our probabilistic sliding mechanism neglects seismic or hydrological landslide triggers (e.g. Keefer, 1984, 2002; Keefer and

Larsen, 2007; Marc et al., 2015, 2018, 2019). Rather we simulate landslides as a stochastic process based on the mechanical stability of slope patches. Future developments could however adjust the spatial probability of landsliding by coupling an explicit earthquake model to HyLands (cfr., Croissant et al., 2019). The probability for co-seismic landslide activity can then be directly obtained by using constrained relationships between Peak Ground Acceleration (PGA) and landslide initiation probabilities (Meunier et al., 2007).

HyLands uses a non-linear, multiple-flow sediment redistribution scheme depending on topographic slope. Accurately simulating landslide sediment run-out distances is however a challenging process which is difficult to constrain and often simulated using empirical approximations (e.g. based on the absolute height difference within a landslide (Claessens et al., 2007)). A potential way to validate and calibrate landslide runout distance would be to compare landslide-derived sediment distributions




simulated with HyLands with runout distances simulated with higher complexity models (Iverson, 2000; Iverson and George,
2016; Zhou et al., 2020). Nevertheless, in its current form, HyLands reproduces characteristics of landscapes and channel pro-
files dominated by deep-seated landslides (Ouimet et al., 2007), and is therefore a useful tool to study the interaction between
river incision and landslide dynamics at landscape evolution space and time scales.

### 4.3  Calibration of HyLands

A main challenge when applying HyLands in real settings is the calibration of both the river incision and landslide parameters.
The power of any LEM lies in its capacity to integrate data over multiple spatial and temporal timescales. Therefore, a range
of datasets can be used to constrain model parameters. We identify three main categories of potential calibration data.

1. Topographic parameters that can be derived from DEMs. These include a range of metrics describing river character-
   istics (drainage density, river steepness, river stream power), hillslope properties (slope distribution, mean and median
   slope angles, aspect), and landslide scaling relationships (magnitude-frequency and area-volume distributions). All these
metrics can be derived from topographic data and subsequently used to constrain HyLands erosion parameters (cfr. Fig.
   7).

2. Data directly constraining the the integrated effects of river incision or landslide erosion. Ongoing efforts to map mass
   movements in landslide prone areas now enable estimates of erosion rates over decal timescales (Hovius et al., 1997;
   Larsen and Montgomery, 2012). Data on sediment redistribution following landslide events is however more difficult to
collect. Although initial compilations now exist of global landslide sediment mobilization rates (Broeckx et al., 2020),
   such inventories remain incomplete. Data on landslide mobilization rates can be used to train HyLands while in turn,
   HyLands can be used to further extend datasets on landslide mobilisation rates and predict landslide sediment production
   rates in regions which are otherwise difficult to access.

3. Catchment-averaged cosmogenic radionuclide (CRN) derived erosion rates. CRN data has been used to calibrate river
incision models that explicitly integrate the stochastic nature of fluvial incision over time (DiBiase and Whipple, 2011;
   Scherler et al., 2017; Campforts et al., 2019). However, CRN data is sensitive to landslide activity (Niemi et al., 2005;
   Yanites et al., 2009; Wang et al., 2017) and the calibration of stochastic river incision models has been shown to be
   sensitive to landslide activity (Campforts et al., 2019). HyLands directly simulates stochastic landsliding and hence
   enables explicit simulation of the impact of landsliding on CRN-derived erosion rates, making them a promising tool to
constrain HyLands.

### 4.4  Potential applications

Given the capacity of HyLands to explicitly simulate the interaction between fluvial dynamics and landslide triggering, it
provides a unique toolbox to advance the field of geomorphology on several fronts. In the following we outline two example
applications.





First, HyLands can be used as an experimental environment to test how landslides influence landscape response to external perturbations. Landslides are known to mediate long term landscape evolution (Korup, 2005; Korup et al., 2007). By altering sediment fluxes, they fundamentally alter the dynamic equilibrium between hillslopes and rivers, resulting in long-term implications for landscape evolution (Egholm et al., 2013). Moreover, large landslides are reported to critically alter drainage networks by causing major river captures (Korup et al., 2007; Dahlquist et al., 2018).

Second, HyLands can be used to evaluate the response time of a landscape to a landslide event and to understand the timescales over which landslide-derived sediments are exported from the landscape (Wang et al., 2015; Li et al., 2016; Yanites et al., 2010). As illustrated in this paper, a landscape requires a certain response time to recover from a landslide event—or a series of landslide events—and to evolve back to a steady-state configuration (Fig. 10). Landslide activity is typically manifested in downstream sediment dynamics and LEMs are the right tool to simulate large-scale landscape response to landslide activity

in upstream mountain regions. HyLands will enable prediction of downstream sediment response to landsliding, provided that the model can be calibrated effectively (section 4.3).

## 5 Conclusions

We presented a new, fully coupled model for river incision into bedrock, sediment transport, and bedrock landsliding. HyLands couples a mass conservative sediment-flux-dependent incision model (SPACE, Shobe et al., 2017) with a deep-seated landslide

algorithm (Densmore et al., 1998) and a multiple-flow sediment redistribution algorithm (Carretier et al., 2018). HyLands is designed to simulate landscape evolution at large temporal and spatial scales. The fluvial component of the model matches known, steady-state analytical solutions developed in earlier work (Davy and Lague, 2009; Shobe et al., 2017). Landslides produced by HyLands replicate observed scaling relationships indicating the realism of the simulations. HyLands is implemented in the TopoToolbox GIS interface (Schwanghart and Kuhn, 2010), thereby facilitating the use of rasterized field data

for calibration and providing direct access to a wide range of GIS analysis tools. In an example application, we illustrated how HyLands can be used to evaluate the impact of landslide activity on fluvial and hillslope characteristics. We showed how landslide activity triggers the formation of landslide-dammed lakes and how HyLands is capable of simulating subsequent lake infilling and knickpoint formation, similar to reported landscape changes following landslide activity (Ouimet et al., 2007). The foremost advantage of HyLands is its capacity to explicitly simulate the role of landslides, landslide-derived sediment

and fluvial dynamics at the landscape scale. The model is well-suited to address a range of new questions related to how channel-hillslope coupling modulates landscape response to external perturbations.

*Code availability.* The HyLands 1.0 TTLEM component as well as all other TTLEM components used in this paper are part of TopoToolbox version 2. The source code and future updates are available in the GIT repository: https://github.com/BCampforts/topotoolbox. The exact version of the software code used to produce the results presented in this paper is archived on Zenodo (https://zenodo.org/badge/

latestdoi/78645261). Upon publication we will submit a pull request to the master repository where TopoToolbox is housed: (wschwanghart/topotoolbox). Documentation, installation instructions, and software dependencies for the HyLands project can be found at https://github.



com/BCampforts/topotoolbox. Detailed scripts and user manuals for the simulations illustrated in this paper can be found at https://github.com/BCampforts/pub_hylands_campforts_etal_GMD which is also archived on Zenodo: https://zenodo.org/badge/latestdoi/247779084. HyLands is platform independent and requires MATLAB 2014b or higher and the Image Processing Toolbox. The HyLands modeling framework

is distributed under a MIT open-source license.

*Data availability.* Digital elevation models used in this paper are derived from the 30 m SRTM v3 DEM (NASA JPL, 2013). The resampled DEM is available through https://github.com/BCampforts/topotoolbox

*Video supplement.* Videos are described in Table 2 of the main text which contains hyperlinks to the following movies: https://doi.org/10.5446/45969; https://doi.org/10.5446/45967; https://doi.org/10.5446/45968; https://doi.org/10.5446/45973; https://doi.org/10.5446/45970;

https://doi.org/10.5446/45971; https://doi.org/10.5446/45972

*Author contributions.* BC and CMS conceived the conceptual model on landslide-derived sediment dynamics and designed the research project. BC developed the algorithm with help from CMS. BC and CMS took the lead in writing the paper. Concepts to verify and evaluate model components were conceptualized by PS, DL, MVM and JB. All authors contributed to shaping the research, analyses, and paper.

*Competing interests.* The authors declare that they have no conflict of interest.

*Acknowledgements.* BC received funding from the Research Foundation Flanders, FWO grant agreement no. 12Z6518N. CMS received funding from the European Union's Horizon 2020 research and innovation programme under the Marie Skłodowska-Curie grant agreement no. 833132. PS and DL have received funding from the European Research Council (ERC) under the European Union's Horizon Horizon 2020 research and innovation programme (grant agreement no. 803721).





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
