# Peer review of "HyLands 1.0: a Hybrid Landscape evolution model to simulate the impact of landslides and landslide-derived sediment on landscape evolution"

_Geoscientific Model Development, 2020_

## Referee Comment (RC1) · Anonymous Referee #1 · 15 Apr 2020

**Summary**:

The authors present a new landscape evolution model, HyLands, that combines models of landsliding and bedrock evolution. The backbone of the model, SPACE by Shobe et al. 2017, can shift between both transport-limited and detachment-limited cases of landscape evolution and therefore simulate the continuum of bedrock to mixed bedrock-alluvial to alluvial rivers. Here, bedrock erosion is modulated by the cover effect, which greatly depends on the rate that sediment is delivered to the channel. HyLands combines SPACE with a landsliding model that allows for sediment delivery in a highly punctuated fashion instead of a steady rate. The authors first demonstrate the steady-state solutions for the SPACE model in the Topo Toolbox

[Figure]

Landscape Evolution Model (TTLEM) framework. They then add landsliding to a natural landscape (Namche-Barwa region) and found that the modeled and observed characteristics of the landslide dynamics match quite well. Last, they devise a model run on a synthetic landscape where there is a 100-year period of intense landsliding to simulate widespread co-seismic landslides. They found that landslides create drainage rerouting from landslide blockage and generate channel knickpoints. They conclude by discussing calibration techniques and potential applications for HyLands.

**Review**:

This manuscript is well written and contains a detailed description of the numerical model, HyLands. The literature review covers the field of numerical landscape evolution modeling and makes a compelling argument for why a model like HyLands is needed. The objectives and motivation are well-thought out and are clear to the reader. The discussion is thorough, and I appreciate the effort the authors took to flesh out potential calibration techniques and applications to their model. They conclude by stating that their model is "well-suited to address a range of new questions related to how channel-hillslope coupling modulates landscape response," which I wholeheartedly believe. However, I think this manuscript should take a more in-depth look at the steady-state behavior of this model. I believe this manuscript should be accepted with some minor revisions.

*Verification*

The manuscript shows steady-state solutions for detachment-limited, transport-limited, and mixed bedrock-alluvial cases. These solutions and the associate figure are quite similar to the work in Shobe et al., 2017, and I am not sure it is totally necessary for them to be repeated in this manuscript. Figure 8 shows HyLands working remarkably well compared to the data of Larsen and Montgomery, 2012, but it seems that the model systematically overestimates landslide volumes for all scales of landslides. The author's attribute the overestimation of small landslide volumes to the inability of the

model to deposit materials in the landslide scars. What is the reasoning for the model overestimating large landslide volumes?

*Synthetic Landscapes*
At what spatial scale is the drainage re-routing occurring? From Figure 9 (d, e, and f) and Figure 10, it does not seem that the channel profile's location has changed significantly. The figures make it seem like the channel moves on the order of one cell size due to valley blockage and the formation of epigenetic gorges. Could these slight reorganizations, over long periods, create major drainage reorganization or river piracy? Related, how computationally expensive is the landslide (non-linear deposition) routing compared to the rest of the model? I'm really excited for researchers to start using this model. I would be interested to know how fast the model runs, and how modifications that complicate or simplify the landsliding component of HyLands would affect the computational efficiency.

The first part of the model verification section details the steady-state behavior of detachment-limited, transport-limited, and mixed bedrock-alluvial landscapes. I would like authors to answer: **How does the steady state behavior of a mixed bedrock-alluvial landscape with landslides as the sediment delivery mechanism compare to a simulation without landslides?** My guess would be that the main controlling parameter would be $t_{LS}$, the return time for landsliding. For very small values of $t_{LS}$, small frequent landslides will dominate; however, there will be little time for the landscape to recover/build up storage of landslide material. In this case, I believe the model would act very similarly to the initial runs in SPACE. For large $t_{LS}$ values, large but seldom landslides dominate. If the landslides are very rare, I think the landscapes will also act similarly to SPACE. In between these two extremes, I think there is potential for the landscape to behave quite differently. Please consider reading Zhang et al., 2018 (The Advective–Diffusive Morphodynamics of Mixed Bedrock–Alluvial Rivers Subjected to Spatiotemporally Varying Sediment Supply),

which also considers the tool effect.

**Line Comments**:
Figure 2: Where is the function, $f(H/H_*)$, I do not think it is defined in the text. I am guessing it is $[1 - exp(-H/H_*)]$ and $exp(-H/H_*)$. Also, shouldn't the function be on the ordinate and the variable $H/H_*$ be on the abscissa?

Line 171: "landslide" not "andslide"

Line 252: citation for the sink filling algorithm?

Figure 4: Not sure if this plot is made from actual data, but it would be interesting to show a similar figure before and after the landslide for visualization.

Table 1: (a) after Synthetic should be a superscript? Also, you may want to draw another line in the table to make it clear that the Pre, LS-Event, and Post columns refer to the Synthetic landscape and not the Namche-Barwa.

Table 2: Same as Table 1, it is not clear that Before intense LS period belongs to the Synthetic runs, instead of the Real DEM run.

Line 349: Why did you choose 20,000 years for the return time? Would this value affect your results? If it is too long, perhaps you would not collect enough data to generate Figure 8.

Figure 7: Should the unit be $m^{0.5}$, not $m$? Would log units be more useful? Also, perhaps switch the locations of E and D so D is on top, which corresponds to the color bar. Are the color bars for the 1st and 2nd column supposed to be different? Also, the figure caption shows the time steps for the 3rd column as 5, 500, 1500,

and 2000 years, but the row titles show different values. Are they supposed to be different? Is so, why? Last, do landslides stop occurring in the simulation because of the absence of uplift?

Figure 8b: I think there are missing symbols in the legend.

Figure 9 (also, Figure 10 and movies): I think the color bar for topography in panels (a), (b), and (c) are incorrect. It should be from 0 to 300 meters, not 0 to 1 meter.

Figure 9 caption: I think there should be more explanation of how epigenetic gorges are formed in the text. I believe the river jumps out of its original channel after being filled by alluvium and is routed on bedrock. How sensitive is this behavior to the algorithm used to fill sinks?

Figure 10: Where is the rerouting? The channel pathway looks the same to me; is there a better way to illustrate the rerouting?

Line 417: Can you show knickpoint generation with a distance upstream vs. slope plot? The knickpoints are very apparent in the movies, but I do not think a series of topographic profiles would show the knickpoint adequately.

Line 419: "Figs." not "Fig. s".

Line 453: Please consider citing Zhang et al., 2018. This paper looks at how varying sediment transport inputs (e.g. from landsliding) affects bedrock erosion with a tools and cover model.

Line 528: I would be very interested if your model can reproduce this.

---

## Referee Comment (RC2) · Alexander Densmore (Referee) · 28 Apr 2020

This is a very well-written manuscript that makes a clear contribution to knowledge. The authors have combined an elegant new fluvial landscape evolution model with an existing approach to modelling bedrock landslides. The result is, to my knowledge, the only modern landscape evolution model that explicitly accounts for bedrock landslides, and that will therefore allow a number of new problems to be addressed. The authors have done a very good job of summarising both the model and some of these potential applications.

I have made some comments and suggestions on the manuscript PDF, which I will

not repeat here. Most of these are minor and relate to clarification of a few points or requests for a little more information. These should be straightforward for the authors to address. The only more substantive questions relate mostly to the figures, especially Figs 7-10. The text and captions don't fully explain what these figures are showing, making it hard for the reader to fully understand the results. The text describes changes in the lateral position of the river system due to landsliding, but I really don't think that Figs 9-10 show this clearly or effectively. As this seems to be one of their main take-home messages about the impact of landsliding on these landscapes, I think that they could perhaps do more to show these changes to the reader. Once these relatively minor issues are addressed, however, then the revised manuscript should be ready for publication.

Please also note the supplement to this comment:
https://www.geosci-model-dev-discuss.net/gmd-2020-74/gmd-2020-74-RC2-supplement.pdf

**Supplement:**

[revised manuscript text omitted]

---

## Editor Comment (EC1) · Andrew Wickert (Editor) · 6 May 2020

Dear Campforts and co-authors,

Both referees have posted broad and positive comments, and therefore, I will not be soliciting input from a third referee. The open discussion period will remain open until 01 June, and I encourage you to submit responses to referee comments before then. This will also allow you the possibility of having some back-and-forth with the referees, and if your responses are satisfactory, allow me to give you an earlier response regarding whether we will invite you to submit a revised version of your manuscript following your responses to referee comments.

Best wishes,

Andy Wickert
* * *

---

## Author Comment (AC1) · 28 May 2020

**Reviewer 1**

**Summary:** The authors present a new landscape evolution model, HyLands, that combines models of landsliding and bedrock evolution. The backbone of the model, SPACE by Shobe et al. 2017, can shift between both transport-limited and detachment-limited cases of landscape evolution and therefore simulate the continuum of bedrock to mixed

bedrock-alluvial to alluvial rivers. Here, bedrock erosion is modulated by the cover effect, which greatly depends on the rate that sediment is delivered to the channel. Hy-Lands combines SPACE with a landsliding model that allows for sediment delivery in a highly punctuated fashion instead of a steady rate. The authors first demonstrate the steady-state solutions for the SPACE model in the Topo Toolbox. Landscape Evolution Model (TTLEM) framework. They then add landsliding to a natural landscape (Namche-Barwa region) and found that the modeled and observed characteristics of the landslide dynamics match quite well. Last, they devise a model run on a synthetic landscape where there is a 100-year period of intense landsliding to simulate widespread co-seismic landslides. They found that landslides create drainage rerouting from landslide blockage and generate channel knickpoints. They conclude by discussing calibration techniques and potential applications for HyLands.

**Review:** This manuscript is well written and contains a detailed description of the numerical model, HyLands. The literature review covers the field of numerical landscape evolution modeling and makes a compelling argument for why a model like HyLands is needed. The objectives and motivation are well-thought out and are clear to the reader. The discussion is thorough, and I appreciate the effort the authors took to flesh out potential calibration techniques and applications to their model. They conclude by stating that their model is "well-suited to address a range of new questions related to how channel-hillslope coupling modulates landscape response," which I wholeheartedly believe. However, I think this manuscript should take a more in-depth look at the steady-state behavior of this model. I believe this manuscript should be accepted with some minor revisions.

*Verification:*

The manuscript shows steady-state solutions for detachment-limited, transport-limited,and mixed bedrock-alluvial cases. These solutions and the associate figure are quite similar to the work in Shobe et al., 2017, and I am not sure it is totally necessary for them to be repeated in this manuscript. Figure 8 shows HyLands working remarkably well compared to the data of Larsen and Montgomery, 2012, but it seems that the model systematically overestimates landslide volumes for all scales of landslides. The author's attribute the overestimation of small landslide volumes to the inability of the model to deposit materials in the landslide scars. What is the reasoning for the model overestimating large landslide volumes?

*Synthetic Landscapes:*

At what spatial scale is the drainage re-routing occurring? From Figure 9 (d, e, and f) and Figure 10, it does not seem that the channel profile's location has changed significantly. The figures make it seem like the channel moves on the order of one cell size due to valley blockage and the formation of epigenetic gorges. Could these slight reorganizations, over long periods, create major drainage reorganization or river piracy? Related, how computationally expensive is the landslide (non-linear deposition) routing compared to the rest of the model? I'm really excited for researchers to start using this model. I would be interested to know how fast the model runs, and how modifications that complicate or simplify the landsliding component of HyLands would affect the computational efficiency.The first part of the model verification section details the steady-state behavior of detachment-limited, transport-limited, and mixed bedrock-alluvial landscapes. I would like authors to answer: **How does the steady state behavior of a mixed bedrock-alluvial landscape with landslides as the sediment delivery mechanism compare to a simulation without landslides?** My guess would be that the main controlling parameter would be $t_{LS}$, the return time for landsliding. For very small values of $t_{LS}$, small frequent landslides will dominate; however, there will be little time for the landscape to recover/build up storage of landslide material. In this case, I believe the model would act very similarly to the initial runs in SPACE. For large $t_{LS}$ values, large but seldom landslides dominate. If the landslides are very rare, I think the landscapes will also act similarly to SPACE. In between these two extremes, I think there is potential for the landscape to behave quite differently. Please consider reading Zhang et al., 2018 (The Advective-Diffusive Morphodynamics of Mixed Bedrock-Alluvial Rivers Subjected to Spatiotemporally Varying Sediment Supply) paper which also considers the tool effect.

**Reply**: We explicitly want to thank reviewer 1, to review our manuscript in such a short period, given the challenging times. We are pleased that the reviewer appreciates our work and agrees on the need for the development of LEMs like HyLands. In the following, we address her/his specific comments.

**RC 1.1** — The manuscript shows steady-state solutions for detachment-limited, transport-limited,and mixed bedrock-alluvial cases. These solutions and the associate figure are quite similar to the work in Shobe et al., 2017, and I am not sure it is totally necessary for them to be repeated in this manuscript.

**Reply**: We indeed reproduced the analytical verification methods for SPACE as earlier proposed by Shobe et al., 2017. We also considered moving this part to a supplementary file but decided to keep in in the main body of the text because of the following. Space has been developed and tested in the Landlab framework. We ported the same set of equations to the TTLEM modelling environment. To validate our implementation we tested it thoroughly by comparing model output with these well established sets of analytical fluvial equations. While this is not new from a scientific point of view, we believe it is important for every new numerical model to be tested rigorously against such benchmark equations. Given the scope of the GMD journal, we therefore decided to report on these comparisons and keep the fluvial model verification exercise in the main text of the manuscript. Moreover, the set of model runs we use here, is used at a later stage in the paper to show the impact of landslides on fluvial sediment dynamics. Showing the functionally of the fluvial component is therefore key to support our findings documented at a later stage in this paper.

**RC 1.2** — Figure 8 shows HyLands working remarkably well compared to the data

of Larsen and Montgomery, 2012, but it seems that the model systematically overesti-mates landslide volumes for all scales of landslides. The author's attribute the overes-timation of small landslide volumes to the inability of the model to deposit materials in the landslide scars. What is the reasoning for the model overestimating large landslide volumes?

**Reply**: This is a valid point carefully observed by the reviewer. As we explicitly mention in section 3.2.2, we use the Namche-Barwa area solely to demonstrate and evaluate the performance of HyLands. We do not take into account a number of boundary conditions (such as uplift patterns, see section 3.2.2) which prevents us to reproduce exact features of landscape exhumation in this region. Therefore, we ran the model with standard parameters values and did not calibrate any of them (see also RC 1.5). We evaluated the the performance of the landslide algorithm against its capability of reproducing the shape of empirical universal magnitude-frequency and area-volume relationships. We did not aim to exactly reproduce the observed scaling relationships since this would involve calibration and uncertainty analysis of the model, which is beyond the scope of this paper. Regarding the Area-Volume relationship in particular, what we see is that the volume of small landslides deviates from the otherwise linear Area-Volume relationship (in a loglog space). Regardless of the carefully observed fact that overall, tis particular model run indeed seems to over predict landslide volumes. To improve the model fit, there are three essential landslide parameters which will adjust landslide volumes : $C$, $phi$ and $t_{LS}$. Calibrating those will be feasible following pathways outlined in section 4.3 of the manuscript. We agree with the reviewer that we could have stated this more clearly and will rephrase some of the sentences in the corresponding paragraph:

- Figure 8.b shows that HyLands is capable of approaching the shape of the uni-versal Area-Volume relationships found by...

- While HyLands seems to overestimate simulated landslide volumes for very small

landslides, the Area-Volume relationship simulated with HyLands approaches a linear relationship for larger landslides, similar to the shape of the observed Area-Volume relationship. Note that overall, landslide volumes as simulated with Hy-Lands are slightly over predicted in comparison to observations. Study area specific model calibration would improve this fit but is beyond the scope of this this model evaluation in which we evaluate the capacity of HyLands to reproduce the shape of the universal area-volume relationship. We attribute the positive deviation from the linear Area-Volume relationship for smaller landslides to the nature of the landslide algorithm ...

**RC 1.3** — At what spatial scale is the drainage re-routing occurring? From Figure 9 (d, e, and f) and Figure 10, it does not seem that the channel profile's location has changed significantly. The figures make it seem like the channel moves on the order of one cell size due to valley blockage and the formation of epigenetic gorges. Could these slight reorganizations, over long periods, create major drainage reorganization or river piracy?

**Reply**: Again, a very insightful comment. Testing the impact of landslides on river capture and drainage reorganisation would be a natural avenue for follow up research activities. The model setup we used to showcase the impact of landslides on landscape evolution does however not provide the 'right' tectonic configuration to test this hypothesis. In our synthetic model run, we focus on the the coupling between landslides and river-bed morphology. We therefore use a model set-up which is similar to the one used to evaluate the fluvial components of HyLands (Space). The initial surface of this run is a tilted plain which drains towards the southwestern corner, the only open boundary node. Therefore, from the first run steps onwards, all the water is forced toward this lower left corner. In order tho test whether the model actually reproduces river captures and drainage organisations, we suggest model setups with

open flow boundary conditions. Moreover, you would probably like to test the impact of uplift or precipitation perturbations, which are, for the sake of simplicity and model demonstration, all kept constant in the current model runs.

**RC 1.4** — Related, how computationally expensive is the landslide (non-linear deposition) routing compared to the rest of the model? I'm really excited for researchers to start using this model. I would be interested to know how fast the model runs, and how modifications that complicate or simplify the landsliding component of HyLands would affect the computational efficiency.

**Reply**: This is a relevant comment which we believe requires some attention given the aim of this paper (i.e. presenting a novel numerical model). The good news is that HyLands is fairly efficient both regarding landslide formation (the Culmann algorithm) as well as the sediment routing algorithm. In the updated version of the manuscript, we will added a row in the Table 1, indicating the average time required to complete one model iteration (Computation time per iteration). From the synthetic model runs, it can be seen that running HyLands with landslide erosion and sediment redistribution takes about double the time as it would when those processes are not simulated.

**RC 1.5** — The first part of the model verification section details the steady-state behavior of detachment-limited, transport-limited, and mixed bedrock-alluvial landscapes. I would like authors to answer: How does the steady state behavior of a mixed bedrock-alluvial landscape with landslides as the sediment delivery mechanism compare to a simulation without landslides?. My guess would be that the main controlling parameter would be $t_{LS}$, the return time for landsliding. For very small values of $t_{LS}$, small frequent landslides will dominate; however, there will be little time for the landscape to recover/build up storage of landslide material. In this case, I believe the model would act very similarly to the initial runs in SPACE. For large $t_{LS}$ values, large but seldom

landslides dominate. If the landslides are very rare, I think the landscapes will also act similarly to SPACE. In between these two extremes, I think there is potential for the landscape to behave quite differently.

**Reply**: Evaluating the impact of landslides on long term landscape evolution is part of the motivation why we developed HyLands. Answering the question as to what extent landslides impact steady state landscape outlooks however opens up a bunch of other questions. A first question is related to the impact of different parameter values on the landslide erosion dynamics: the reviewer is right in his assessment that landslide return times $t_{LS}$ will impact steady state landscape topography. However, equally important will be the cohesion factor $C$ as well as the angle of internal friction $\phi$. The way in which these factors influence landslide erosion patterns is currently not well understood. HyLands offers a tool to investigate these inter-dependencies using a suit of sensitivity analyses and by comparing simulated landslide patterns with observed landslide properties. Second, also the way in which landslide sediments are being distributed will influence 'steady state' landscape shapes. Again, running the model using a broad range of parameter values will improve our understanding as to what extend sediment redistribution influences landscape evolution. Parameters involved here are those controlling landslide sediment deposition on hillslopes after failure (Eq. 12, parameter $Sc$) as well as subsequent sediment redistribution by fluvial processes (the SPACE parameters). Finally, the way in which landscapes evolve towards a steady state will be at least as important to evaluate as the steady state result of landscape evolution. Answers to all those questions are currently open for debate. Nevertheless, we believe that this manuscript is not the right place to answer them: we want to use this paper to present a novel model and to evaluate its basic functionality. Albeit showing the results of one particular model run where a landscape is evolving to steady state might answer some of the previous questions, the answer would be a partial one given the strong interdependency of all processes involved in HyLands. Understanding those interdependencies in a rigorous sensitivity analysis would be a first step in

answering the question as to what extent landslides influence long term landscape dynamics. The reason we did run the model into a steady state without landslides (Fig. 5) is because we wanted to test if our model is capable to reproduce well established theoretical relationships on fluvial dynamics which currently do not exist for landslides. We believe however that the reviewer proposes a very interesting potential application of HyLands which we now address in the discussion section of the manuscript (under 4.4: Potential applications) where we added the following paragraph:

A particular question which remains open for debate is the way in which landslides influence the evolution of a landscape to steady state. Albeit the stochastic nature of landslides will prevent landscapes to evolve towards time and space invariant topographies, even with landslides, landscapes will evolve towards a quasi steady sate if external drivers such as climate and tectonics remain constant. Although our mechanistic understanding of landscapes strongly improved by studying steady state landscapes, an even more interesting and challenging question would be to study the impact of landslides on the dynamic evolution of a landscape towards such a steady state. The latter being more relevant for most real-world landscapes which are known to be rather in transient than a steady state (Mudd et al. 2017).

**RC 1.6** — Please consider reading Zhang et al., 2018 (The Advective-Diffusive Morphodynamics of Mixed Bedrock-Alluvial Rivers Subjected to Spatiotemporally Varying Sediment Supply) paper which also considers the tool effect.

**Reply**: We thank the reviewer for pointing us to this paper. Shobe et al (2017) discussed similarities and differences between the SPACE model and the approach of Zhang et al (2018). We agree that the tools effect would be another interesting addition to the current model framework.

Minor

**Reviewer Point 1.7** — Figure 2: Where is the function, $f(H/H\star)$, I do not think it is defined in the text. I am guessing it is $(1 - exp(\frac{-H}{H\star}))$ and $exp(\frac{-H}{H\star})$. Also, shouldn't the function be on the ordinate and the variable $H/H\star$ be on the absciss a?

**Reply**: The reviewer is correct about the form of the function $f(H/H\star)$. It is a good point that although these expressions occur in equations 3 and 4, $f(H/H\star)$ was never explicitly defined on its own. We have added its definition to the caption of Figure 2, and in the same place referenced the relevant governing equations (3 and 4).

Both reviewers commented on the choice of axes in this figure. We have reversed the ordinate and abcissa.

**Reviewer Point 1.8** — Line 171: "landslide" not "andslide"

**Reply**: Fixed.

**Reviewer Point 1.9** — Line 252: citation for the sink filling algorithm?

**Reply**: Fixed.

**Reviewer Point 1.10** — Figure 4: Not sure if this plot is made from actual data, but it would be interesting to show a similar figure before and after the landslide for visualization.

**Reply**: This is a hypothetical sketch. Adjusted the subscript by adding 'potentially initiate'

**Reviewer Point 1.11** — Table 1: (a) after Synthetic should be a superscript?Also, you may want to draw another line in the table to make it clear that the Pre, LS-Event, and Post columns refer to the Synthetic landscape and not the Namche-Barwa.

**Reply**: Fixed.

**Reviewer Point 1.12** — Table 2: Same as Table 1, it is not clear that Before intense LS period belongs to the Synthetic runs, instead of the Real DEM run.

**Reply**: Fixed.

**Reviewer Point 1.13** — Line 349: Why did you choose 20,000 years for the return time?Would this value affect your results? If it is too long, perhaps you would not collect enough data to generate Figure 8.

**Reply**: Good question. We did not calibrate any of the model parameters for reasons discussed in the manuscript and in RP 1.2. Although the other parameters could be set to theoretical values, $t_L S$ is a new parameter introduced in this model. We therefore set the $t_L S$ to $2 \times 10^4$ years which is a rather arbitrary value. Parameter sensitivity runs in future work will show the impact of changing the landslide return times. We added the following sentence in the manuscript to clarify: Evaluation of model sensitivity to changing values for $t_{LS}$ would be one of the natural avenues for further work.

**Reviewer Point 1.14** — Figure 7: Should the unit be m$^{0.5}$, not m? Would log units be more useful?Also, perhaps switch the locations of E and D so D is on top, which corresponds to the color bar. Are the color bars for the 1st and 2nd column supposed to be different?Also, the figure caption shows the time steps for the 3rd column as 5, 500, 1500, and 2000 years, but the row titles show different values. Are they supposed to

be different? Is so, why? Last, do landslides stop occurring in the simulation because of the absence of uplift?

**Reply**: All very good suggestions. Fixed to $m^{0.5}$. I definitely tried log units because those would be more familiar to the reader. Unfortunately, this does not really work out well since small erosion and deposition rates would end up being negative (values smaller than 1). This would prevent us from plotting erosion and deposition on the same plot. Color bars are the same, and since patterns of landslides are almost not different in the previous version of this figure, we dropped the colorbars. There was an error in the caption. We removed this part of the caption as the years are already indicated in the first sentence of the caption. Landslides do not stop to occur. This is more clear on the new version of the figure. Given that the second reviewer also had some valuable suggestions for this figure, we made a new version of Figure 7.

**Reviewer Point 1.15** — Figure 8b: I think there are missing symbols in the legend.

**Reply**: Sorry for that, we messed up the legend. The grey bar is now properly added to the figure.

**Reviewer Point 1.16** — Figure 9 (also, Figure 10 and movies): I think the color bar for topography in panels (a), (b), and (c) are incorrect. It should be from 0 to 300 meters, not 0 to 1meter.

**Reply**: You are absolutely right. We corrected this in both the figures and the movies. Moreover, we made several additional adjustments to this figure in order to improve clarity and to get the message better across (see also SP **??**)

**Reviewer Point 1.17** — Figure 9 caption: I think there should be more explanation of how epigenetic gorges are formed in the text. I believe the river jumps out of its original

channel after being filled by alluvium and is routed on bedrock. How sensitive is this behavior to the algorithm used to fill sinks?

**Reply**: We rephrased the corresponding paragraph in the text as:
The drainage re-routing mechanism dominates in the simulations presented here and results in the formation of epigenetic river gorges (Fig. 9). Epigenetic river gorges are characterized by rivers incising into the bedrock of former valley walls due to the blockage of the formal channel by landslide derived sediment (Ouimet et al. 2008).
Regarding the sensitivity to the fill algorithm: after landslide blockage of the river path, a fill algorithm is used to identify landslide lakes and water is rerouted following the steepest path using a D8 flow direction algorithm.

**Reviewer Point 1.18** — Figure 10: Where is the rerouting? The channel pathway looks the same to me; is there a better way to illustrate the rerouting?

**Reply**: The rerouting happens on Figure 9, when landsliding kicks in. A major rerouting happens right after the start of the LS simulations (Fig 9.c to the LS Fig.9, d). Small changes to the flow path continue to occur from Fig. 9.d to f. Once landsliding stops, the channels are not blocked any longer and will mostly stay in place (Fig. 10)

**Reviewer Point 1.19** — Line 417: Can you show knickpoint generation with a distance upstream vs.slope plot? The knickpoints are very apparent in the movies, but I do not think a series of topographic profiles would show the knickpoint adequately.

**Reply**: We do not fully understand this comment. We believe the presence of knick-points is very apparent on Fig. 9 d-f. We added a reference to this figure in the manuscript to enhance clarity. We also added some text to better explain the phenom-ena of epigenetic river gorges.
**Reviewer Point 1.20** — Line 419: "Figs." not "Fig. s"

**Reply**: Thanks. Fixed.

**Reviewer Point 1.21** — Line 453: Please consider citing Zhang et al., 2018. This paper looks at how varying sediment transport inputs (e.g. from landsliding) affects bedrock erosion with a tools and cover model.

**Reply**: Done, see also comment before.

**Reviewer Point 1.22** — Line 528: I would be very interested if your model can reproduce this.

**Reply**: We too.

———————————————————

---

## Author Comment (AC2) · 28 May 2020

**Reviewer 1**

See AC1

**Reviewer 2**

**Alexander Densmore**

**Summary:** This is a very well-written manuscript that makes a clear contribution to knowledge.The authors have combined an elegant new fluvial landscape evolution model with an existing approach to modelling bedrock landslides. The result is, to my knowledge, the only modern landscape evolution model that explicitly accounts for bedrock landslides,and that will therefore allow a number of new problems to be addressed. The authors have done a very good job of summarising both the model and some of these potential applications. I have made some comments and suggestions on the manuscript PDF, which I will paper not repeat here. Most of these are minor and relate to clarification of a few points or requests for a little more information. These should be straightforward for the authors to address. The only more substantive questions relate mostly to the figures,especially Figs 7-10. The text and captions don't fully explain what these figures are showing, making it hard for the reader to fully understand the results. The text describes changes in the lateral position of the river system due to landsliding, but I really don't think that Figs 9-10 show this clearly or effectively. As this seems to be one of their main take-home messages about the impact of landsliding on these landscapes, I think that they could perhaps do more to show these changes to the reader. Once these relatively minor issues are addressed, however, then the revised manuscript should be ready for publication.

**Reply**: We explicitly want to thank the reviewer, Alexander Densmore, to review our manuscript in such a short period, given the challenging times. We are pleased that the reviewer appreciates our work. Minor comments regarding typos and text edits are addressed directly in the updated version of the manuscript.

**Reviewer Point 2.1** — Line 6 - remove earth

**Reply**:  Fixed.

**Reviewer Point 2.2**  —  Line 64 - This isn't actually the case - I had to go back and check! We used the lowest point on a hillslope that fit the failure criteria, but that point did not need to be in the channel. As stated on p. 15,208, 'This ensures that landslides begin near the toes of hillslopes', but not necessarily at the toe. Line 66 - As above, this isn't what was done in that paper, so I suggest cutting this. You're absolutely right that sediment is spread at a constant slope and that there's absolutely nothing mechanistic about the approach, however.

**Reply**:  Thanks for clarifying this and apologies for misinterpreting this. We removed this sentence and rephrased to: (i) all hillslopes behave as Mohr-Coulomb materials (Taylor et al. 1948), (ii) landslides initialize near the toes of hillslopes and (iii) landslide-derived sediment is spread under a constant slope, following the steepest downslope path.

**Reviewer Point 2.3**  —  LLine 76 - The wording here is a little confusing - it sounds like the processes aren't available at large scales, which isn't what you mean.  I suggest rewording as something like '...processes, and require input parameters which may not be adequately known at large spatial scales.'

**Reply**:  Good suggestion, we rephrased accordingly.

**Reviewer Point 2.4**  —  Line 81 - While the text above is very clear on what has been done to date, I feel like there is a sentence missing that just puts those pieces together into a single statement that motivates your work. In other words: what's the specific gap that you will now be able to fill?

**Reply**: We added the following sentence: Notwithstanding the prominent role of landslides in shaping the earth surface and controlling sediment supply and transport, few efforts have been made to actively simulate the impact of stochastic landsliding on landscape evolution and sediment dynamics over large spatial and temporal scales.

**Reviewer Point 2.5** — Line 107 - due to landsliding

**Reply**: Fixed.

**Reviewer Point 2.6** — Line 112 - This was already defined on line 58

**Reply**: Fixed.

**Reviewer Point 2.7** — Line 116 - OK... with the caveat that this is also going to depend upon the spatial resolution of the model and the way in which rivers are modelled in the grid - i.e., whether or not they are treated as a single thread of cells, or whether the equations are applied to the whole landscape. I presume it's the latter although this isn't explicitly stated

**Reply**: We added two sentences to the previous paragraph for clarification: Note that HyLands does not explicitly distinguish between river or hillslope cells: all equations are applied to the entire landscape. Processes affecting sediment thickness and bedrock elevation in each cell can be either fluvial dynamics (SPACE), landslides, or a combination of both, hence the hybrid nature of HyLands. Moreover, as suggested by A. Densmore, we moved the following sentence from the discussion to this point in the manuscript: Note that this approach implies that all river cells in the landscape are assumed to occupy 1 grid cell with distance $dx$, that channel width may be less

than, equal to, or greater than $dx$, and that river width is only a function of contributing drainage area.

**Reviewer Point 2.8** — Line 135 - the

**Reply**: Fixed.

**Reviewer Point 2.9** — Line 155 - Can you remind us (briefly) how this is determined?

**Reply**: We added the following sentences to clarify: V is the net effective settling velocity, which represents the still-water particle settling velocity corrected for the upward effects of turbulence and the vertical gradient in sediment concentration through the water column (Davy and Lague, 2009). HyLands enables spatially variable values for V to distinguish between settling velocities over flooded versus non-flooded nodes.

**Reviewer Point 2.10** — Figure 2: There is a slight mismatch with the text here, given that the text doesn't refer to f at all, but simply builds negative exponential functions of H/H* into eqns 3 and 4. I wonder, therefore, if it's more straightforward to flip this by 90 deg and to relate this more clearly to eqns 3 and 4. I get the echoes here of the tools/cover effect plots, but I think it's potentially a bit confusing as currently designed. Just a thought.

**Reply**: Good point. We flipped the axes as suggested, and defined $f\left(H/H_*\right)$ in the figure caption. We also referenced the relevant erosion/entrainment equations in the caption to make the function notation less confusing.

**Reviewer Point 2.11** — Line 207 - plane

**Reply**:  Fixed.

**Reviewer Point 2.12**  —  Line 208 You use both node and pixel in this section - are they equivalent?  If so then I suggest using one term or the other; if not then please explain the distinction.

**Reply**:  Good point. We use the term 'cell' now throughout the text

**Reviewer Point 2.13**  —  Line 217 - Suspended sediment makes sense here - I'm struggling to envision a situation, however, where a measurable volumetric fraction of hillslope sediment contributes instantly to the dissolved load of the river. Perhaps cut, unless I'm missing something?

**Reply**: We dropped dissolved

**Reviewer Point 2.14**  —  Line 222 - True - and also doesn't account for different depositional slopes for different landslide bulk rheologies or grain size distributions...

**Reply**: Thanks for clarifying

**Reviewer Point 2.15**  —  Line 230 - an approach

**Reply**:  Fixed.

**Reviewer Point 2.16**  —  Line 232 - landslide-derived

**Reply**:  Fixed.
**Reviewer Point 2.17** — Line 245 - ... and there is no deposition at that cell?

**Reply**: We added these words for clarification.

**Reviewer Point 2.18** — Line 246 - Is this the angle of the surface of the resulting deposit? If so, then maybe call it a minimal deposit surface angle. 'Spreading angle' could be confused with spreading across multiple flow directions.

**Reply**: Good suggestion. We adjusted the text accordingly throughout the manuscript

**Reviewer Point 2.19** — Line 249 – Should this be changed to 'over the landscape'? Presumably the spreading algorithm distributes sediment downslope, whether or not the target cell is a hillslope or channel cell. This comes back to an earlier question - is there any distinction made between hillslope and channel cells, or are the model equations applied to the whole landscape? The previous text suggests the latter, but this sentence might imply that there is a difference. It would be great if you could clarify this.

**Reply**: Good that you point us to this. We added a couple of sentences right after the GMB equation (Eq. 5) to clarify. See also reply to earlier comment (2.7)

**Reviewer Point 2.20** — cfr. == cf. ?

**Reply**: Fixed

**Reviewer Point 2.21** — Line 251 - Again - are nodes and cells the same thing? If so then it would be good to use a consistent term.

[Figure]

**Reply**: Good point. We use the term 'cell' now throughout the text

**Reviewer Point 2.22** — Line - 272 conditions?

**Reply**: Fixed

**Reviewer Point 2.23** — Table 1 - It took me awhile to realise that (a) referred to a note at the bottom of the table - perhaps make this superscript to match the others?

**Reply**: Fixed

**Reviewer Point 2.24** — Table 1 - I suggest inserting a space (m $yr^{-1}$), to avoid confusion.

**Reply**: Fixed

**Reviewer Point 2.25** — Table 1 - This should be mentioned explicitly in section 2, rather than defined in the table notes

**Reply**: We added this information in the main text of the manuscript, after introducing Eq. 5, see also reply to earlier comment (2.9)

**Reviewer Point 2.26** — Line 310 - Applying HyLands to the Namche Barwa-Gyala Peri massif I don't have any issue with this application... but I find it a slightly odd choice, not least because of the very limited field data available to ground-truth the Larsen and Montgomery landslide inventory. Given the rapidly-growing number of

well-constrained inventories out there, why did you choose this particular one? The pre-1974 inventory is particularly poorly constrained in terms of the time scale that it covers, and both inventories suffer from extreme orthorectification issues caused by the steep topography. There's also nothing known about the history of either rainfall or earthquake landslide triggers in that area, other than the big 1950 event which almost certainly triggered some of the events in the inventory. It's not a bad choice to evaluate the model, but it just seems like there are other inventories out there that fit your requirements better. I'd be curious to see an additional line in the text that gives the reason why this was chosen.

**Reply**: Again, a very insightful comment. We agree with the reviewer that if the aim of this exercise would be to exactly reconstruct an observed LS inventory, other regions would probably make up for a better application for reasons given by the reviewer. However, our intention is not to calibrate HyLands to a specific study area, neither to reproduce exact magnitude frequency distributions because these would indeed require detailed information on earthquake and storm histories. Rather we were interested if we could reproduce the general shape of the empirical and universally observed magnitude frequency and area-volume relationships. The question remains as to why we selected the Namche Barwa-Gyala Peri massif as an area to test HyLands. We now address this issue in the manuscript by adding the following lines of text:

We selected the Namche Barwa-Gyala Peri massif to evaluate the performance of HyLands given its unique geomorphologic configuration featuring amongst the highest globally documented river stream power in combination with very active hillslope processes (Larsen, 2012). With HyLands being designed to couple the role of fluvial and hillslope processes, this region makes up for a good test environment. Note however that we do not intent to calibrate neither validate the model but run it using fixed, theoretical model parameters (section 3.2.3). Applications of HyLands aiming to constrain the model through parameter calibration and validation (section 4.3) would require additional data to ground-truth landslide inventories and to provide detailed records on

landslide triggers such as earthquakes and storms.
Also note the reply to Reviewer 1, which is related to this comment (see AC1 RC 1.2)

**Reviewer Point 2.27** — Line 320 - Out of curiosity, why would you do this?

**Reply**: We address this question in the manuscript now: We resampled the DEM to a resolution of 20 m in order to evaluate the capacity of HyLands to reproduce the rollover in the magnitude frequency distribution, often reported to occur for landslide areas $< 900$ m$^2$, which would be the minimum landslide area when using the original SRTM data.

**Reviewer Point 2.28** — Line 342 - It's not clear what this text is doing within the citation - perhaps rework this into the sentence.

**Reply**: Fixed

**Reviewer Point 2.29** — Figure 7 - It's quite hard to see the detail in this figure without zooming way in - I wonder if you can make more efficient use of the space by increasing the size of the panels. Given that the colorbars for each row are almost identical, do they need to be shown 4 times?

**Reply**: Good suggestion, we remade the figure.

**Reviewer Point 2.30** — Figure 7 - I might be missing something, but the left-hand column just seems to show landslide locations - I can't see anything that follows the red-to-blue color scale indicated. The other two colorbars seem to fit with the middle

and right-hand columns, but what are the colors meant to indicate on the left-hand column?

**Reply**: Good point. Actually in the left hand column of the previous figure, you can see the landslides if you would zoom in closely. However, as this is very difficult to see, we removed the colorbar for these figures.

**Reviewer Point 2.31** — Figure 7 - It's not very clear what you're plotting. All of your model parameters relating to erosion and deposition are represented as rates, with units of L/T. So it's not obvious why you've taken the square root of those quantities and how you've kept units of meters. I understand that this won't affect the patterns that you show, but I think this could be more clear to the reader.

**Reply**: This remark is similar to the one made by reviewer 1. We corrected the units to $\sqrt{m}$. See also see also AC1 SP1.14

**Reviewer Point 2.32** — Figure 7 - Rather than referring to this as 'SED' in the figure, it would be better to relate this back to the parameters that you have already defined and used throughout the manuscript so far. Is this the same as $H$ in equation 1?

**Reply**: Yes, this is $H$, we changed the label of the colorbar.

**Reviewer Point 2.33** — Line - 360 I'm not sure where that can be seen on Fig 7 - perhaps point it out?

**Reply**: Good suggestion. We now point it out explicitly: e.g., the deposition pattern in Fig. 7.h reflects the shape of erosion patterns resulting from previous landslide activity

**Reviewer Point 2.34** — Figure 8 - Rather than 'PDF', it might be better to label this for what it is, which is the spatial frequency density of landsliding per unit area

**Reply**: Good suggestion, we will adjust

**Reviewer Point 2.35** — Figure 8 - This may be a problem with the PDF conversion, but the symbol for this zone seems to be missing from the legend on the figure, along with the best-fit regression line

**Reply**: Sorry for that. We messed up the legend of the figure, this is fixed now

**Reviewer Point 2.36** — Figure 9 - The caption for this figure is a little bit lacking, in that it's not clear what is being plotted. What's the difference between the top-left and top-right subfigure in each panel? What does the blue line in each panel represent? Why is the brown line labelled 'Current Topo' in the left-hand column, but seems to correspond to 'Sediment' on the right-hand y-axis? The brown lines seem to show different things in the two columns, so I'd suggest making these distinct. Also, confusingly, blue areas seem to denote sediment on the profiles in the left-hand column, but water in the profiles on the right - I didn't realise that this was the case until I got to Fig. 10 a couple of pages later. This is a really interesting figure - a little more care with the colors, labels, and caption would really help the reader to get the most out of it.

**Reply**: Thanks a lot for these very useful recommendations. We adjusted the labels on the figure and changed the figure caption as follows: **(a-d)** Time slices showing evolution of the landscape to steady state, before the landslide period. The upper left subplots show the evolution of topography through time. The upper right subplots show the evolution of of sediment thickness ($H$) through time. On both subplots, the blue line represents the location of the river, plotted in the lower subplots. These lower

subplots show the topographic and bedrock elevation (red and black line respectively). The difference between the topographic elevation and the elevation of the bedrock represents the sediment thickness. With respect to total elevation, sediment thickness is small, which is why sediment thickness (orange line) is also plotted against a separate right-hand y-axis. The gray shaded area represents bedrock underlying the river profile. **(e-h)** Time slices showing the landslide period where intense landsliding is occurring over a period of 100 years. The upper left subplots show the landslide activity. The location of landslides is indicated with black diamonds. The colors represent the square root of the landslide erosion (-) and deposition (+) during the presented time step. The upper right subplots show the evolution of of sediment thickness ($H$) through time. On both subplots, the blue line represents the location of the river, plotted in the lower subplots. These lower subplots show the topographic and bedrock elevation (red and black line respectively) as well as the volume occupied by sediments and water (orange and blue shaded area respectively). Note that, during landsliding, both pure landslide dams arise as well as irregularities in the bedrock profile (the grey bumps). The latter originate from the river being redirected after landsliding forming epigenetic gorges (see text). We adjusted Figure 10 accordingly.

**Reviewer Point 2.37** — Line 402 - I don't understand - does this mean that the profile is always taken in the same place, but that in some places that profile corresponds to the active channel and in other places it doesn't (when the channel has been diverted to a different location)? Or are those bumps areas where bedrock incision and lowering of the channel bed has been inhibited by the addition of large volumes of sediment?

**Reply**: We agree that this was a confusing sentence and removed it from the manuscript. Instead, we now elaborate on this issue in the next paragraph by extending our explanation on the formation of epigenitic gorges. This comment is similar to the remark of reviewer 1, addressed in AC1 RC1.3 and one of the following remarks (PT 2.39)

**Reviewer Point 2.38** — Line 407 - Just to clarify, landslide sediment has the same transport coefficient as any other sediment in the model, right? So there is no 'immobile debris'?

**Reply**: Correct, we removed immobile

**Reviewer Point 2.39** — Line 412 - I'm not sure that I would call that 'drainage re-routing', as that implies a lateral shift in the position of the active channel. Is that what you mean?

**Reply**: We actually mean to describe such a lateral shift. We rephrased the corresponding paragraph in the text as:
The drainage re-routing mechanism dominates in the simulations presented here and results in the formation of epigenetic river gorges (Fig. 10). Epigenetic river gorges are characterized by rivers incising into the bedrock of former valley walls due to the blockage of the formal channel by landslide derived sediment (Ouimet et al. 2008).

**Reviewer Point 2.40** — Line 457 - See my earlier queries on section 2 - this information could usefully be included there.

**Reply**: Good suggestion, we move this sentence to section 2. See also reply SP 2.7 .

**Reviewer Point 2.41** — Line 494 - True... or even with medium-complexity approaches such as RAMMS or Flow-R...

**Reply**: Indeed, we added those and corresponding references

**Reviewer Point 2.42** — Line 509 - True. You could cite Fan et al. (2018) Landslides as an example where this has been done, or Fan et al. (2019) Rev of Geophys as a good review of the problem.

**Reply**: Absolutely, a reference to the review of Fan et al. was intended here, good that you point us this

**Reviewer Point 2.43** — Line 519 - them: Not sure what you're referring to here.

**Reply**: We adjusted the sentence

**Reviewer Point 2.44** — Line 526 - OK - so, given the results of those studies, as well as the recent work by Thomas Croissant as well as some of the authors, what are the most pressing remaining questions or issues?

**Reply**: One example of a pressing remaining question has been suggested by reviewer 1 and is now added as a potential application to this paragraph (see also AC1 RC1.5): A particular question which remains open for debate is the way in which landslides influence the evolution of a landscape to steady state. Albeit the stochastic nature of landslides will prevent landscapes to evolve towards time and space invariant topographies, even with landslides, landscapes will evolve towards a quasi steady sate if external drivers such as climate and tectonics remain constant. Although our mechanistic understanding of landscapes strongly improved by studying steady state landscapes, an even more interesting and challenging question would be to study the impact of landslides on the dynamic evolution of a landscape towards such a steady state. The latter being more relevant for most real-world landscapes which are known to be rather in transient than a steady state (Mudd et al. 2017).

**Reviewer Point 2.45** — Line 530 - ... or to a major landslide triggering event. See, for example, some of the work after the 2015 Gorkha earthquake that speculated on this exact point.

**Reply**: Good suggestion. We rephrased and inserted some additional references as: Second, HyLands can be used to evaluate the response time of a landscape to a major landslide triggering event and to understand the timescales over which landslide-derived sediments are exported from the landscape (Wang et al., 2015; Li et al.,2016; Schwanghart et al., 2016; Robinson et al., 2017; Roback et al., 2018)

---

## Author Comment (AC3) · 28 May 2020

We want to thank the editor, Andy Wickert, for his swift follow up. We have posted a reply to RC1 and RC2 on the discussion forum.

---

## Author Response (AR2)

**Response to the editor**

We thank the editor for his thoughtful and constructive review of our manuscript. In the following we address his concerns and suggestions point by point.
* * *
**Summary:** The authors responded appropriately and thoroughly to the positive reviews, so I decided not to send the revised version back out to the referees. To ensure that a final set of eyes other than the authors passed over the article, I have now read it and reviewed it thoroughly. Almost all of my comments are minor – sometimes to the point of copyediting – aside from the need for a user's manual and a question about porosity in landslide deposits.

Overall, I find this to be a very exciting piece of work. I am glad to see an explicit treatment of mechanistically based sediment production in an LEM, and conservation of that sediment through the remainder of the system. This is a good add-on to the SPACE work by Shobe et al., and a major dose of realism beyond the stream-power-plus-hillslope-diffusion simple equations that typically are used to simulate landscape change over time.

Following the authors' treatment of these minor concerns, I believe that the article will be fit to publish as a strong contribution to landscape-evolution modeling.

**Editor Point 1.1** — User manual: This could not be found at [https://github.com/BCampforts/pub_hylands_campforts_etal_GMD](https://github.com/BCampforts/pub_hylands_campforts_etal_GMD) or the (broken) links on that page, and must be completed prior to publication.

**Reply**: Thanks for checking this. The links were broken because I changed the default branch to the HyLands one. Everything should be fixed now. We also copied the installation instructions for this version of HyLands on this repo and refer the readers to the main repo of HyLands for updates.

**Editor Point 1.2** — eq. 1: consider using text for fluv, sed, hill

**Reply**: fixed

**Editor Point 1.3** — 161-162: water discharge per unit width

**Reply**: fixed

**Editor Point 1.4** — 180. Perhaps add a sentence as to why the failure plane bisects these two angles, to inform readers who do not understand why the failure should not just happen at the angle of internal friction. (I do not know this intuitively either, but guess that it may relate to the distribution of mass above the plan in the wedge.)

**Reply**: We clarified Note that Eq. 6 implies that failure occurs at angles higher than $\phi$, being the consequence of the force balance controlled by material properties (Eq. 8) and the mass of the wedge above the failure plane (Culmann, 1875).

**Editor Point 1.5** — Eq. 9: perhaps give "rnd" a single-letter definition; multiple-letter variables can look like multiplication or functions.

**Reply**: Fixed

**Editor Point 1.6** — 231. I think that few beyond me care about this, but a "volumetric flux" should have units of $length^3/time/length^2$, with the cross-sectional area normalization making it a flux. Volumetric discharge is $L^3/t$. This is so routinely ignored that it probably doesn't matter to use the terms right, as much as it pains me.

**Reply**: Good point and we agree. We changed to volumetric sediment discharge here, in Eq.5 and throughout the text

**Editor Point 1.7** — 230-231. Is this for the full hillslope area or just for the landsliding portion of the slopes? The discrepancy between the sub-sub-section title and the next sentence has me a bit confused.

**Reply**: This applies to the for the full hillslope area. We changed the subsection title to "Hillslope-derived sediment" to clarify

**Editor Point 1.8** — 235. as suspended sediment –¿ in high suspension. (And perhaps replace "instantaneously" with "rapidly". I think that your key point is that you want the sediment that still interacts with the bed. Because you use "sediment" here in both senses (all produced by the hillslope, that which you care about), I suggest that you simplify.

**Reply**: Fixed

**Editor Point 1.9** — 238. This section leaves me wondering about whether you take into account porosity. Landslide material – especially from bedrock – may increase in volume after it is released due to the addition of pore spaces. It seems like a reasonable idea to have this be the same value that you use for river-bed sediments, in order to not have to deal with varying porosities all across the landscape. Of course, adding porosity will create the problem that you have more volume of material to deposit than was eroded. Hopefully this can be implemented without too much trouble.

**Reply**: Luckily we did, we assume the same porosity for hillslope derived sediment as for other fluvial sediments (given that we only have one sediment layer). In the code this is done in e.g. the $D_{Hill}$ function at line 49 of `https://github.com/BCampforts/topotoolbox/blob/HyLands/ttlem/hylands/D_Hill.m`. We clarified by adding the following sentence Note that both $E_{s_{hill}}$ and $D_{s_{hill}}$ are corrected for dimensionless sediment porosity ($\phi_{sed}$ in Eq. 1) thereby factoring in the increase in bedrock derived landslide sediment volumes during deposition due to the addition of pore spaces.

**Editor Point 1.10** — 239. This paragraph could use a topic sentence.

**Reply**: We added: In the following we describe how deposition of landslide derived material is being calculated.

**Editor Point 1.11** — 250. which –¿ that

**Reply**: fixed

**Editor Point 1.12** — 263. text, not code :) So $\infty$ not $inf$ to use the symbol

**Reply**: fixed

**Editor Point 1.13** — 264-266. grammar/clarity

**Reply**: We rephrased as: A minimal angle ($\delta$) under which landslide-derived sediments should be deposited can be set as a parameter value but is not required to run HyLands

**Editor Point 1.14** — 273. Perhaps note $V_{Lake}$ here (and/or refer to the table)

**Reply**: Good suggestion. We inserted : HyLands enables spatially variable values for $V$ (Eq. 5) to distinguish between settling velocities in non-flooded versus flooded cells by changing the values for $V$ and $V_{Lake}$, respectively (see Table 1).

**Editor Point 1.15** — 281. detachment –¿ detachment-limited

**Reply**: fixed

**Editor Point 1.16** — 300. Since this is an issue of choices made within SPACE, and not something germane to HyLands, no need to address. But I would point out that sediment input equaling U * A implies that there is absolutely no clast breakdown! This is not internally consistent with your term to remove the fine fraction from landslide inputs. There are a few relevant articles that describe this issue: * Attal et al. (2015): Impact of change in erosion rate and landscape steepness on hillslope and fluvial sediments grain size in the Feather River basin (Sierra Nevada, California) * Sklar et al. (2017): The problem of predicting the size distribution of sediment supplied by hillslopes to rivers * Wickert and Schildgen (2019): Long-profile evolution of transport-limited gravel-bed rivers Importantly here, we (WS) show that a linear $Q_{sin} = U * A$ relationship predicts channels with long profiles that are straight to convex. There's nothing that you have to do here to address this point, but I thought it might be important to be aware of it, and the more general way of thinking of your "fine fraction removal" from landslides. Furthermore, the two hillslope papers (above) give some useful insights in to the grain-size distribution reaching the channel, which might be of interest to you.

**Reply**: Interesting comments, and thanks for pointing us to these relevant papers. The statement $Q_s = UA$ only applies in our work as a steady-state condition used to close the SPACE analytical solutions (approach inherited from Davy and Lague 2009) in the absence of any landsliding. However, we do acknowledge the simplicity of this approach and the inherent assumption about clast breakdown, especially once landslide-derived sediments are introduced. On a related note, the current Eq. 15 reflects the simplified case of $F_{f_{fluv}} = 0$ so we changed the text to reflect that more clearly.

**Editor Point 1.17** — 365. Assuming bed sed porosity = 0 makes sense for this test (and may help you to address my comment on line 238), but neglecting an easy multiplier that adds realism doesn't make much sense to me. Still – proof of concept – you're okay in my book.

**Reply**: Sediment porosity is taken into account in HyLands, both for fluvial and hillslope sediment, just not for this test (see my previous reply).

**Editor Point 1.18** — 505. A more recent reference for river width and sediment supply is: Pfeiffer et al. (2017), Sediment supply controls equilibrium channel geometry in gravel rivers. There are also good papers on alluvial rivers by Colin Phillips and Kieran Dunne, but these are probably less relevant to your high-relief and rapidly uplifting landscape. (The Phillips paper is: "Self-organization of river channels as a critical filter on climate signals".)

**Reply**: The paper of Pfeiffer et al. is really to the point here, we included the reference

**Editor Point 1.19** — 559-565. An excellent reference for this would be Tofelde et al. (2018): Effects of deep-seated versus shallow hillslope processes on cosmogenic 10Be concentrations in fluvial sand and gravel. (Please feel no pressure to include it, though: I'm a coauthor, so would usually avoid noting the reference out of concern over conflict of interest, except that in this case it is so precisely geared towards your point.)

**Reply**: Agreed, very relevant paper at this point, we included the reference

[revised manuscript text omitted]
_{\mathrm{fluv}}} + \left( \frac{D_{s_{\mathrm{fluv}}} - E_{s_{\mathrm{fluv}}}}{1 - \phi_{sed}} \right) \\
& - E_{r_{\mathrm{hill}}} + \left( \frac{D_{s_{\mathrm{hill}}} - E_{s_{\mathrm{hill}}}}{1 - \phi_{sed}} \right)
\end{aligned}
\tag{1}
$$

where $\eta$ $[L]$ is the topographic elevation given by the sum of the bedrock elevation $R$ $[L]$ and the bed sediment thickness $H$ $[L]$. $U$ $[L/T]$ is the rock uplift rate and $\phi_{sed}$ is the  sediment porosity. $E_{r_{\mathrm{fluv}}}$ $[L/T]$ is the fluvial volumetric erosion flux of bedrock per unit bed area, representing the amount of bedrock that is detached and entrained into the water column. $E_{s_{\mathrm{fluv}}}$ $[L/T]$ is the fluvial volumetric entrainment flux of sediment per unit bed area and $D_{s_{\mathrm{fluv}}}$ $[L/T]$ is the fluvial volumetric deposition flux of sediment per unit bed area. $E_{r_{\mathrm{hill}}}$ $[L/T]$ is the volumetric flux of hillslope bedrock erosion due to landslding per unit bed area, representing the amount of bedrock that is detached. $E_{s_{\mathrm{hill}}}$ $[L/T]$ is the volumetric entrainment flux of sediment erosion (produced by landsliding or creep) per unit bed area and $D_{s_{\mathrm{hill}}}$ $[L/T]$ is the volumetric deposition flux of hillslope-derived sediment per unit bed area. Note that HyLands does not explicitly distinguish between river or hillslope cells: all equations are applied to the entire landscape. Processes affecting sediment thickness and bedrock elevation in each cell can be either fluvial dynamics (SPACE), landslides, or a combination of both, hence the hybrid nature of HyLands.

**2.1 River sediment transport and bedrock erosion**

HyLands uses the SPACE river erosion model of Shobe et al. (2017). SPACE has two key advantages for the purposes of modeling river response to landslide sediment delivery. First, because of its derivation from the erosion-deposition family of models (e.g., Beaumont et al., 1992; Davy and Lague, 2009), it can dynamically shift between detachment-limited (erosion is limited by the rate of sediment or bedrock detachment from the bed) and transport-limited (erosion is limited by the capacity

of the flow to move detached sediment) behavior. Second, it can simulate the full continuum of possible river bed compositions from bare bedrock channels to mixed bedrock-alluvial channels to fully alluvial channels. This is accomplished by combining mass conservation of river bed sediment with a bedrock erosion law to simultaneously solve for the time evolution of the bedrock and sediment surfaces. Note that this approach implies that all river cells in the landscape are assumed to occupy 1 grid cell of width $dx$, that channel width may be less than, equal to, or greater than $dx$, and that river width is only a function of contributing drainage area. We implement the SPACE model equations in the TTLEM MATLAB modeling framework. For a full overview of the SPACE model and comparison with other models for coupled sediment and bedrock channel evolution, see Shobe et al. (2017).

**2.1.1 Fluvial sediment and rock mass conservation**

Conservation of sediment closely follows the erosion-deposition approach of Davy and Lague (2009), with the addition of terms that represent the entrainment of detached bedrock in the water column (Fig. 1). The spatial change in volumetric sediment  discharge $Q_{s_{fluv}}$ $\left[L^3/T\right]$ per unit width $w$ $[L]$ is written as:

$$\underline{\frac{\partial\left(Q_{sfluv}/w\right)}{\partial x}}\frac{\partial\left(Q_{s_{fluv}}/w\right)}{\partial x} = E_{s_{fluv}\,s_{fluv}} + \left(1 - F_{f_{fluv}\,f_{fluv}}\right)E_{r_{fluv}\,r_{fluv}} - D_{s_{fluv}\,
[revised manuscript text omitted]

---

## Author Response (AR3)

We updated the location of the reference to Tofelde et al. as suggested by the editor.